# Understanding and Mitigating Miscalibration in Prompt Tuning for Vision-Language Models

**Shuoyuan Wang** [1]  **Yixuan Li** [2]  **Hongxin Wei** [1]

## Abstract

Confidence calibration is critical for the safe deployment of machine learning models in the real world. However, such issue in vision-language models like CLIP, particularly after fine-tuning, has not been fully addressed. In this work, we demonstrate that existing prompt tuning methods usually lead to a trade-off of calibration between base and new classes: the cross-entropy loss used in standard prompt tuning (e.g., CoOp) causes overconfidence in new classes by increasing textual label divergence, whereas regularization-based tuning (e.g., KgCoOp) maintains the confidence level but results in underconfidence in base classes due to the improved accuracy. Inspired by the observations, we introduce Dynamic Outlier Regularization (DOR) to ensure the confidence calibration on both base and new classes after fine-tuning. In particular, DOR minimizes the feature deviation of novel textual labels (instead of base classes) sampled from a large vocabulary set. In effect, DOR prevents the increase in textual divergence for new labels while easing restrictions on base classes. Extensive experiments demonstrate that DOR can notably enhance the calibration performance of current fine-tuning methods. Our code is available at https://github.com/ml-stat-Sustech/Outlier-Calibration.

## 1. Introduction

Large pre-trained vision-language models (VLMs) like CLIP (Radford et al., 2021) have become the de facto standard in today's zero-shot tasks including image recognition (Wortsman et al., 2022), open-vocabulary segmen-

tation (Liang et al., 2023) and knowledge-augmented retrieval (Ming & Li, 2024). To efficiently adapt CLIP to domain-specific downstream tasks, various parameter-efficient fine-tuning (PEFT) techniques, such as prompt tuning (Zhou et al., 2022b) and adapter (Gao et al., 2024), have been proposed. While these methods improve accuracy, they largely overlook reliability concerns such as confidence calibration in fine-tuned CLIP. Without fully understanding the miscalibration in fine-tuned CLIP, it can exacerbate safety concerns in high-stakes applications like medical diagnosis and autonomous driving.

In confidence calibration, we generally expect the model's confidence level to be consistent with its empirical accuracy (Tu et al., 2023). In the literature, zero-shot CLIP is often recognized for its excellent performance in confidence calibration (Minderer et al., 2021). Prior work (Wang et al., 2024) has shown that fine-tuning CLIP on base classes often leads to miscalibration on novel classes within the same task, where the model is expected to generalize (Zhou et al., 2022b; Yao et al., 2023). While these studies focus on miscalibration in novel classes unseen during fine-tuning, they do not adequately address calibration issues on base classes. Overall, the community still lacks a comprehensive understanding of the fundamental cause of miscalibration in fine-tuned CLIP and effective strategies for mitigation.

In this work, we first investigate how existing prompt tuning methods affect the confidence calibration of CLIP. In general, these methods can be fell into two categories: standard fine-tuning (e.g., CoOp (Zhou et al., 2022b)) and regularization-based fine-tuning (e.g., KgCoOp (Yao et al., 2023)). Our empirical analysis first reveals that both approaches struggle to maintain calibration across base and novel classes simultaneously, inevitably compromising one: *CoOp exhibits overconfidence on new classes while KgCoOp provides underconfident predictions on base classes.* To thoroughly understand such a phenomenon, we explain it from the perspective of textual feature divergence. In particular, CoOp increases the divergence of textual label distribution through cross-entropy loss, resulting in excessively high confidence misaligned with actual accuracy on new classes. Conversely, KgCoOp constrains the divergence increase, preserving confidence levels across both base and

[1]Department of Statistics and Data Science, Southern University of Science and Technology, Shenzhen, China. [2]Department of Computer Sciences, University of Wisconsin-Madison, WI, USA. Correspondence to: Hongxin Wei <weihx@sustech.edu.cn>.

*Proceedings of the $42^{nd}$ International Conference on Machine Learning*, Vancouver, Canada. PMLR 267, 2025. Copyright 2025 by the author(s).

novel classes. However, this leads to the underconfidence issue due to the improved accuracy on base classes. This raises a key question: ***Can we achieve reliable calibration on both base and novel classes after fine-tuning?***

To tackle the above challenges, we introduce **D**ynamic **O**utlier **R**egularization (**DOR**), a regularization technique incorporated into the fine-tuning phase. The core idea of DOR is to leverage outliers to regulate the divergence of unseen textual distribution in the fine-tuned CLIP, without interfering with the vanilla fine-tuning objectives. Specifically, we first collect textual outliers from the large lexical database (e.g., WordNet (Miller, 1995)), while ensuring that the selected words do not overlap with the base classes in the fine-tuning task. During fine-tuning, we minimize the feature discrepancy of novel textual labels between the fine-tuned model and the zero-shot CLIP. To further enhance flexibility, the textual outliers are dynamically sampled from the constructed set in each epoch. By incorporating dynamic textual outliers, DOR mitigates the increase in textual divergence for novel labels while reducing constraints on base classes, improving overall calibration stability.

We verify the effectiveness of DOR across 11 image classification datasets and 4 types of ImageNets with covariant shifts. Extensive experiments show that DOR can enhance the calibration of existing prompt-tuning methods (see Table 1). For instance, DOR reduces the Expected Calibration Error (ECE) by an average of 8.09% for CoOp across the 11 downstream datasets. Moreover, our method can maintain and even improve the generalization performance of those tuning methods (See Table 6). DOR also achieves competitive improvements in the presence of covariate shifts, further validating its robustness. Beyond prompt-based tuning, we show that this regularization criterion can be extended to visual fine-tuning methods using image outliers, highlighting its broader applicability.

We summarize our main contributions as follows:

1. We show that current prompt-tuning methods typically lead to a trade-off between base and new classes, compromising one of them. To understand such miscalibration, we provide an in-depth analysis from the perspective of textual distribution divergence.

2. We propose DOR, a simple yet effective regularization that ensures calibration performance on both base and new classes. Our method is compatible with existing prompt-tuning methods and can be extended to visual fine-tuning methods with image outliers.

3. Extensive experiments on a wide range of real-world benchmarks demonstrate that DOR is effective and easy-to-use. DOR can be easily incorporated with various prompt tuning methods.

## 2. Preliminaries

**Contrastive Language-Image Pretraining (CLIP)**  CLIP is a visual-language model that measures the alignment between images and texts (Radford et al., 2021). Recently, CLIP has shown great potential in zero-shot inference for arbitrary classes. Let $\phi : \boldsymbol{x} \to \mathbb{R}^d$ and $\psi : \boldsymbol{t} \to \mathbb{R}^d$ denote CLIP's image and text encoders, respectively. Given an image instance $\boldsymbol{x}$ and a text label $c$, the logit function of CLIP can be formulated as:

$$L_c^{clip}\left(\boldsymbol{x}_i\right) = \tau \cdot \text{sim}\left(\phi(\boldsymbol{x}), \psi(\boldsymbol{t}_c)\right). \tag{1}$$

Here $\boldsymbol{t}_c$ is derived from a hand-crafted prompt like "a photo of a {class}", where the "{class}" is filled with the text label $c$. $\tau$ is generally set as a pre-trained constant of 100.

For multi-class classification, we predict by selecting the label with the highest probabilities among the label candidate set $\mathcal{C} = \{c_i\}_{i=1}^C$, as shown below:

$$c^* = \arg\max_{c \in \mathcal{C}} p(c|\boldsymbol{x}) = \arg\max_{c \in \mathcal{C}} \frac{e^{L_c^{clip}(\boldsymbol{x})}}{\sum_{i=1}^C e^{L_i^{clip}(\boldsymbol{x})}} \tag{2}$$

where $p(c|\boldsymbol{x})$ is the predicted probability of class $c$ for the instance $\boldsymbol{x}$.

**Prompt tuning**  To boost performance of CLIP in downstream tasks, prompt tuning methods have been proposed to efficiently fine-tune CLIP on datasets of interest (Zhou et al., 2022a;b). In particular, prompt tuning optimizes the context prompt only, without retraining the model and updating its weights. CoOp (Zhou et al., 2022b) replaces the hand-crafted textual tokens with a set of learnable textual token $\mathcal{T} = \{\boldsymbol{v}_1, \boldsymbol{v}_2, \ldots, \boldsymbol{v}_M\}$, where $M$ is the length of tokens. Thus, the output of fine-tuned CLIP is: $L_c^{coop}\left(\boldsymbol{x}\right) = \tau \cdot \text{sim}\left(\phi(\boldsymbol{x}), \psi(\boldsymbol{t}_c')\right)$, where $\boldsymbol{t}_c' = [\boldsymbol{v}_1, \boldsymbol{v}_2, \boldsymbol{v}_3, ... \boldsymbol{v}_M, \boldsymbol{c}]$ and $\boldsymbol{c}$ denotes the textual embedding of class $c$. Using a few labeled samples $\mathcal{D}_{\text{ft}} = \{(\boldsymbol{x}_i, c_i)\}_{i=1}^N$, the learnable textual tokens $\mathcal{T}$ are optimized to minimize the standard cross-entropy (CE) loss $\ell_{ce}$. We refer to the classes used in fine-tuning as *base* classes, and the remaining labels within the same task as *new* or *novel* classes.

To enhance the generalization ability of the learnable prompt for unseen classes within the task, *regularization-based* methods like KgCoOp (Yao et al., 2023) introduces a regularization term to align the learned prompt to the hand-crafted prompt. The optimization of KgCoOp is:

$$\mathcal{T}^\star = \arg\min_{\mathcal{T}} \left\{ \frac{1}{N} \sum_{i=1}^N \ell_{ce}\left(p\left(c_i \mid \boldsymbol{x}_i\right)\right) \right. \\ \left. + \lambda \cdot \frac{1}{C} \sum_{c=1}^C \text{sim}\left(\psi\left(\boldsymbol{t}_c'\right), \psi\left(\boldsymbol{t}_c\right)\right) \right\}. \tag{3}$$

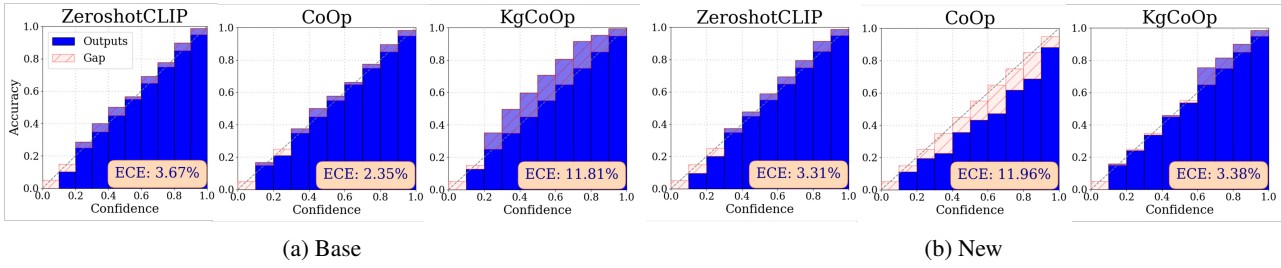

*Figure 1.* Reliability diagram of fine-tuned CLIP (ViT-B/16) on StanfordCars dataset. ECE: Expected Calibration Error (lower is better). Miscalibration is depicted in pink for overconfidence and purple for underconfidence.

Here, the first term is the standard cross-entropy loss used in CoOp and the hyperparameter $\lambda$ is used to control the weight of regularization. With $\lambda = 0$, KgCoOp is degraded to the original CoOp.

**Confidence calibration** In addition to predictive performance, it is generally expected for deep models to be well calibrated, i.e., the predicted class probabilities can faithfully estimate the true probabilities of correctness (Guo et al., 2017). To quantify miscalibration, the *Expected Calibration Error* (ECE) (Guo et al., 2017) is defined as the difference between accuracy and confidence. With $N$ samples grouped into $G$ bins $\{b_1, b_2, \ldots, b_G\}$, the ECE is formulated as:

$$\text{ECE} = \sum_{g=1}^{G} \frac{|b_g|}{N} \left| \text{acc}(b_g) - \text{conf}(b_g) \right|, \quad (4)$$

where $\text{acc}(\cdot)$ and $\text{conf}(\cdot)$ denotes the average accuracy and confidence in bin $b_m$. In the literature, it has been shown that pre-trained CLIP archives excellent performance of confidence calibration in zero-shot inference (Minderer et al., 2021). However, prior work (Wang et al., 2024) finds that fined-tuned CLIP generally suffers from miscalibration on novel classes within the same task, where the model is expected to generalize (Zhou et al., 2022b; Yao et al., 2023). Yet to date, the community still has a limited understanding of the fundamental cause and mitigation strategies of miscalibration during fine-tuning. We proceed by analyzing how the fine-tuning of CLIP affects the calibration.

## 3. Motivation

### 3.1. Empirical study on CLIP calibration

**Setup** To show the miscalibration in fine-tuned CLIP, we first empirically study the calibration performance of fine-tuned VLMs. We use ViT-B-16 pre-trained by OpenAI (Radford et al., 2021) as the zero-shot classification model. In particular, we compare the zero-shot CLIP with standard prompt tuning CoOp (Zhou et al., 2022b) and regularization-based method KgCoOp (Yao et al., 2023) on StanfordCars dataset (Krause et al., 2013). We evaluate the fine-tuned CLIP under *base-to-new* protocol: the dataset is split into

base and new classes. The model is trained only on a few examples from the base classes and evaluated on examples from both base and new classes.

**Prompt tuning leads to a trade-off between base and new classes.** Figure 1 illustrates the calibration performance of zero-shot CLIP, CoOp and KgCoOp on base and new classes. The results show that zero-shot CLIP achieves almost perfect calibration on all classes, while the fine-tuned models cannot maintain the calibration on base and new classes simultaneously, compromising one of them. In particular, CoOp maintains the excellent calibration on base classes but exhibits overconfidence on new classes. Instead, KgCoOp provides underconfident predictions on base classes while preserving the calibration on new classes. This motivates us to further investigate the fundamental cause of miscalibration occurring after fine-tuning.

### 3.2. Understanding the miscalibration in CLIP

Given the above observation, we investigate how prompt tuning leads to the miscalibration issue. Since the visual features remain unchanged in the prompt tuning, the textual features thus play a key role in confidence calibration. We first introduce a feature divergence score to quantify the textual feature variation, inspired by the KNN-based metrics commonly used for distribution estimation in calibration (Xiong et al., 2023; Yuksekgonul et al., 2023).

**Definition 3.1** (Feature Divergence). Consider a feature set $\mathcal{Z} = \{\mathbf{z}_i\}_{i=1}^{N}$, each feature $\mathbf{z}_i \in \mathbb{R}^d$ is embedded by the modality encoder in CLIP. Feature divergence (FD) score of $\mathbf{z}_i$ measures the average distances from each feature to its $M$ nearest neighbors in the set.

$$s_i = \frac{1}{M} \sum_{\mathbf{z}_j \in \mathcal{N}_M(\mathbf{z})} \text{dist}(\mathbf{z}_i, \mathbf{z}_j), \quad (5)$$

where $\mathcal{N}_M(\mathbf{z}_i)$ denotes the set of $M$ nearest neighbors of $\mathbf{z}_i$ and $\text{dist}(\cdot, \cdot)$ is a distance metric like cosine similarity. By averaging these similarity scores across all features, we obtain the overall FD score of a given feature set $\text{FD}(\mathcal{Z}) = \frac{1}{N} \sum_{i=1}^{N} s_i$, which can represent the divergence of the textual distribution. To investigate the miscalibration

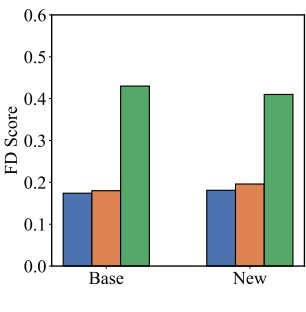
(a) FD score

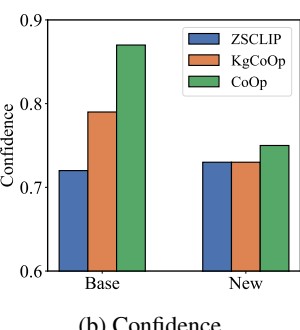
(b) Confidence

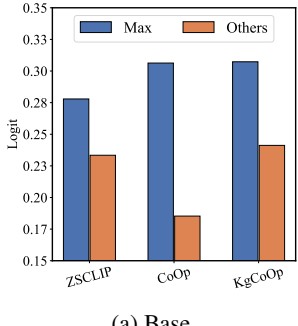
(a) Base

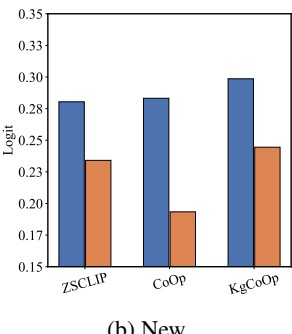
(b) New

*Figure 2.* Results of zero-shot and fined-tuned CLIPs with different prompt tuning methods on UCF101 dataset.

*Figure 3.* Comparison between the maximum logit and the average of other logits, using different prompt tuning methods on DTD.

issue, we vary the hyperparameter $\lambda$ in Eq. (3). In KgCoOp, $\lambda$ is set to 8.0, and it degenerates to CoOp when $\lambda = 0$. We conduct the experiments on UCF101 (Soomro et al., 2012). We present the results in Figure 2.

**CoOp leads to overconfidence on new classes by increasing the textual divergence.** In the analysis of Subsection 3.1 and Figure 2b, we show that CLIP tuned by CoOp exhibits overconfidence on new classes, but keeps calibration on base classes. This is caused by the *CE loss*, which maximizes the posterior $p(y \mid \boldsymbol{x})$ for the ground-truth label $y$ and minimizes the probability for other labels. In other words, CE loss tends to enlarge the distance between the image feature and all textual features except the ground truth. As the image feature remains unchanged during fine-tuning, it can be translated to large distances among all textual labels, including base and new classes. This is supported by the results presented in Figure 2a, which shows that CoOp significantly increases the FD score of textual features compared to zero-shot CLIP. Consequently, As shown in Figure 3, *the gap between the maximum logit and the others widens for both base and new classes*. Hence, the tuned CLIP with CE loss will make softmax predictions with high confidence on both base and new classes. It aligns with the improved accuracy on base classes, but is not consistent with the nearly unchanged accuracy on new classes. This explains why CLIP tuned by CoOp tends to be overconfident on new classes.

**KgCoOp anchors the confidence level by hindering the increase of textual divergence.** In the previous analysis, KgCoOp leads to underconfident predictions on base classes while preserving the calibration on new classes. In Figure 2a, we illustrate the FD scores of textual labels from the zero-shot CLIP and the tuned CLIP by KgCoOp with various $\lambda$. The results show that a large value of $\lambda$ reduces the FD score of both base and new classes, approaching that of zero-shot CLIP. This phenomenon indicates that the regularization in KgCoOp can prevent the model from increasing the textual divergence caused by *CE loss*. Correspondingly, the fine-tuned CLIP by KgCoOp preserves the same confidence level as the zero-shot CLIP. However, the fine-tuning sub-

stantially improves the accuracy of CLIP on base classes, resulting in the underconfidence issue due to the anchored confidence level. In this way, we explain why KgCoOp leads to underconfidence on base classes.

Through the perspective of textual divergence, we provide a thorough explanation for the calibration trade-off caused by different prompt-tuning methods. We provide the comprehensive empirical results on more prompt tuning methods to thoroughly support our observationin Appendix B.1 (See Figure 5 and 6). In addition, we present a *theoretical justification* for the relationship between textual divergence and model confidence in Appendix C. Ideally, we expect to maintain the excellent zero-shot calibration for both base and new classes after fine-tuning. In the following, we proceed by introducing our method, targeting this problem.

## 4. Method: Dynamic Outlier Regularization

In the previous analysis, we show that textual divergence is the key for CLIP calibration. To preserve the calibration capacity of zero-shot CLIP, our key idea is to regularize the textual divergence of new classes without restricting that of base classes. To this end, we use textual outliers to regularize fine-tuned CLIP. We provide clear evidence to show that the regularization using textual outliers can easily mitigate the overconfidence in the new classes without altering the fine-tuning objective in Appendix B.2.

**Selecting textual outliers** With this in mind, we construct a set of textual outliers using nouns from the large language lexical database. Specifically, we mainly use Word-Net (Miller, 1995) as the database, which is a large English lexical database containing over 150,000 words. We select nouns from WordNet that do not overlap but share higher-level concept relations with the base classes used in the fine-tuning, and then incorporate them into our regularization term for prompt tuning. We demonstrate the effectiveness of using relevant textual outliers in Table 8.

Let $\mathcal{C}_{\mathrm{ft}} = \{c_1, c_2, \ldots, c_n\}$ be the $n$ base classes used in

the fine-tuning. First, we obtain a candidate set $\mathcal{C}_{\text{word}} = \{o_1, o_2, \ldots, o_m\}$, by filtering out the base classes from WordNet. Then, we rank the nouns according to the average semantic similarity among the candidate $\mathcal{C}_{\text{word}}$ and each base class $c_j \in \mathcal{C}_{\text{ft}}$. For candidate word $o_i$, we use zero-shot CLIP to quantify the semantic similarity $s_i$:

$$s_i = \frac{1}{n} \sum_{j=1}^{n} \text{sim}\left(\psi\left(\boldsymbol{t}_{o_i}\right), \psi\left(\boldsymbol{t}_{c_j}\right)\right), \qquad (6)$$

where $\boldsymbol{t}_{o_i}$ represents the textual tokens of a noun $o_i$ using a fixed prompt like "a photo of a [class-name]", and $i \in \{1, 2, \ldots, m\}$. $\psi$ is the text encoder of zero-shot CLIP. Then we get the set of textual outliers using the score ranking.

$$\mathcal{O}_{\text{out}} = \{o_i \mid i \in \text{TopK}\left(s_1, s_2, \ldots, s_m\right)\}, \qquad (7)$$

where $\text{TopK}(\cdot)$ represents selecting the indices with the top $K$ largest scores for nouns in the candidate set $\mathcal{C}_{\text{word}}$.

**Dynamic Outlier Regularization** Given a fine-tuning dataset $\mathcal{D}_{\text{ft}}$ with base classes $\mathcal{C}_{\text{ft}}$, we construct a large set of textual outliers $\mathcal{O}_{\text{out}}$ as described above. To prevent the increase of textual divergence on new classes, we propose *Dynamic Outlier Regularization* (**DOR**), which minimizes the feature discrepancy of textual outliers between the zero-shot CLIP and the fine-tuned CLIP.

In each iteration, we randomly sample a batch of textual outliers from the constructed set $\mathcal{O}_{\text{out}}$. We denote the batch of textual outliers as $\mathcal{B} = \{o_i\}_{b=1}^{B}$, where $B$ is the number of textual outliers in the batch. By default, we set $B$ as the same as the batch size of fine-tuning data. Then, we build the regularization by aligning the textual features to those of zero-shot CLIP. Formally, the regularization is defined as:

$$\mathcal{L}_{\text{dor}} = 1 - \frac{1}{B} \sum_{b=1}^{B} \text{sim}\left(\psi\left(\boldsymbol{t}'_{o_b}\right), \psi\left(\boldsymbol{t}_{o_b}\right)\right), \qquad (8)$$

where $\text{sim}(\cdot)$ denotes the cosine similarity function and $\boldsymbol{t}_{o_b}$ denotes the token of the textual outlier $o_b$. Using the regularization, the textual divergence of the fine-tuned CLIP will be regularized to be consistent with the zero-shot CLIP. Different from KgCoOp (Yao et al., 2023), the outlier regularization does not restrict the textual feature deviation of base classes, which is explicitly shown in Figure 9.

Equipped with dynamic outlier regularization, the final training objective for fine-tuning CLIP is:

$$\mathcal{L}_{total} = \mathcal{L}_{\text{ce}} + \lambda \cdot \mathcal{L}_{\text{dor}}, \qquad (9)$$

where $\mathcal{L}_{\text{ce}}$ and $\mathcal{L}_{\text{dor}}$ are the cross-entropy loss of fine-tuning data and the proposed regularization, respectively. $\lambda$ denotes the hyperparameter that controls the weight of the proposed regularization. Our method will degrade to CoOp (Zhou

et al., 2022b) when $\lambda = 0$. As $\lambda$ increases, the optimization will encourage the model to maintain the confidence level on new classes, alleviating the overconfidence issue. We illustrate the effect of $\lambda$ in Figure 4.

**Extension to other robust fine-tuning algorithms** It is worth noting that our regularization is a general method and can be easily incorporated into existing regularization-based tuning for CLIP, including knowledge-guided fine-tuning (Yao et al., 2023; 2024), multimodal consistency (Khattak et al., 2023b; Roy & Etemad, 2023), Self-Regularization (Khattak et al., 2023b) etc. Given the existing robust fine-tuning objective $\mathcal{L}_{\text{robust}}$, we formalize the fine-tuning objective as:

$$\mathcal{L}_{\text{total}} = \mathcal{L}_{\text{robust}} + \lambda \cdot \mathcal{L}_{\text{dor}}. \qquad (10)$$

Noticeably, DOR offers several compelling advantages:

- **Easy-to-use**: DOR leverages text outliers for regularization, which is readily available and easy to collect.

- **Algorithm-agnostic**: DOR can be easily incorporated into existing fine-tuning methods (See Table 1 and 2) or calibration algorithms for CLIP (See Appendix I). Furthermore, our method can be extended to visual fine-tuning methods with image outliers (See Table 7).

- **Fine-tuning-nontoxic**: Compared with existing regularization-based methods, DOR does not conflict with the fine-tuning objective and breaks the calibration trade-off. (See Table 3 and Appendix B.2).

## 5. Experiments

### 5.1. Experimental Setup

**Benchmark setting.** Following recent works (Zhou et al., 2022b; Wang et al., 2024), we perform two evaluations in two standard benchmark settings: 1) *Generalization from Base-to-New Classes*: A downstream dataset will be equally split into base and new classes. The model is trained only on the base classes in a few-shot setting and evaluated on base and new classes. In practice, we mainly focus on the harmonic mean value from both classes. 2) *Domain Generalization*: The model is trained on ImageNet-1k in a few-shot manner and evaluated on four other ImageNet datasets that contain various types of domain shifts.

**Datasets.** For the base-to-new evaluation, we cover diverse classification tasks including ImageNet (Deng et al., 2009), Caltech101 (Fei-Fei et al., 2004), OxfordPets (Parkhi et al., 2012), StanfordCars (Krause et al., 2013), Flowers102 (Nilsback & Zisserman, 2008), Food101 (Bossard et al., 2014), FGVCAircraft (Maji et al., 2013), SUN397 (Xiao et al., 2010), UCF101 (Soomro et al., 2012), DTD (Cimpoi et al., 2014) and EuroSAT (Helber et al., 2019). For domain generalization, we use ImageNet-1k as the source

*Table 1.* Average calibration across 11 datasets. "+DOR(Ours)" to our method applied to standard tuning methods. ↓ indicates smaller values are better. Calibration error is given by $\times 10^{-2}$. "HM" denotes the harmonic mean. **Bold** numbers are *significantly* superior results.

| Method | ECE(↓) Base | New | HM | ACE(↓) Base | New | HM | MCE(↓) Base | New | HM | PIECE(↓) Base | New | HM |
|---|---|---|---|---|---|---|---|---|---|---|---|---|
| ZSCLIP | 3.58 | 4.61 | 4.10 | 3.62 | 4.58 | 4.10 | 0.97 | 1.21 | 1.09 | 6.35 | 6.55 | 6.45 |
| CoOp | 3.07 | 14.58 | 8.82 | 2.97 | 14.50 | 8.73 | 1.07 | 3.72 | 2.40 | 4.68 | 15.27 | 9.98 |
| **+DOR(Ours)** | **2.67** | **6.49** | **4.58** | **2.64** | **6.47** | **4.55** | **0.83** | **1.65** | **1.24** | **4.45** | **8.33** | **6.39** |
| CoCoOp | 3.60 | 6.14 | 4.87 | 3.53 | 6.08 | 4.81 | 0.96 | 1.72 | 1.34 | 5.53 | 7.86 | 6.70 |
| **+DOR(Ours)** | 4.22 | **4.02** | **4.12** | 4.30 | **3.94** | **4.12** | 1.07 | **1.11** | **1.09** | 6.00 | **6.41** | **6.20** |
| MaPLe | 2.75 | 5.46 | 4.11 | 2.65 | 5.42 | 4.04 | 0.82 | 1.52 | 1.17 | 4.71 | 7.37 | 6.04 |
| **+DOR(Ours)** | 2.83 | **4.44** | **3.63** | 2.86 | **4.33** | **3.60** | 0.81 | **1.29** | **1.05** | 4.86 | **6.39** | **5.62** |
| DEPT | 6.04 | 14.58 | 10.31 | 6.00 | 14.52 | 10.26 | 1.44 | 4.58 | 3.01 | 7.31 | 15.42 | 11.37 |
| **+DOR(Ours)** | 7.67 | **7.50** | **7.58** | 7.66 | **7.44** | **7.55** | 1.73 | **1.87** | **1.80** | 8.68 | **8.86** | **8.77** |

*Table 2.* Average ECE (%) of regularization-based methods across 11 datasets. **Bold** numbers are *significantly* superior results. DOR constantly improves the calibration when incorporated with existing regularization-based methods.

| Metric | KgCoOp Vanilla | +DOR | TCP Vanilla | +DOR | PromptSRC Vanilla | +DOR | CoPrompt Vanilla | +DOR | PromptKD Vanilla | +DOR |
|---|---|---|---|---|---|---|---|---|---|---|
| Base | 5.82 | 6.07 | 4.71 | 4.79 | 3.7527 | 3.88 | 2.56 | 2.96 | 4.73 | 4.81 |
| New | 4.48 | **3.99** | 4.07 | **3.80** | 4.15 | **3.80** | 5.96 | **4.69** | 4.38 | **3.66** |
| HM | 5.15 | **5.03** | 4.39 | **4.29** | 3.95 | **3.84** | 4.26 | **3.83** | 4.56 | **4.24** |

dataset and its four variants as target datasets including Im-ageNetV2 (Recht et al., 2019), ImageNet-Sketch (Wang et al., 2019), ImageNet-A (Hendrycks et al., 2021b), and ImageNet-R (Hendrycks et al., 2021a).

**Baselines.** We compare our method with standard prompt tuning algorithms including CoOp (Zhou et al., 2022b), Co-CoOp (Zhou et al., 2022a), MaPLe (Khattak et al., 2023a) and DEPT (Zhang et al., 2024b). We also incorporate DOR with regularization-based tuning including KgCoOp (Yao et al., 2023), TCP (Yao et al., 2024), PromptSRC (Khattak et al., 2023b), CoPrompt (Roy & Etemad, 2023) and PromptKD (Li et al., 2024).

**Implementation details.** We use CLIP ( ViT-B/16) (Rad-ford et al., 2021) as the pre-trained VLM throughout our experiments and report results averaged over 3 runs. We fine-tune the model with 16 samples per class in a few-shot setting (Zhou et al., 2022a). We list the details of the compared methods in Appendix D.

**Evaluation metrics.** We use 4 standard metrics of confidence calibration in our evaluation including Expected Cali-bration Error (ECE) (Guo et al., 2017), Maximum Calibra-tion Error (MCE) (Guo et al., 2017), Adaptive Calibration Error (ACE) (Nixon et al., 2019) and Proximity-Informed Expected Calibration Error (PIECE) (Xiong et al., 2023).

### 5.2. Results

**DOR enhances the calibration of existing prompt-tuning methods.** Table 1 shows the calibration performance of 4 standard tuning baselines w/ or w/o our DOR. We find that DOR can consistently reduce the calibration error on new classes. For instance, DOR significantly reduces the ECE from 14.58% to 6.49% on new classes and maintains the ECE from 3.07% to 2.67% on base classes, which makes CoOp more reliable in terms of predicted confidence. More-over, we incorporate DOR with other robust tuning methods in Table 2. We observe a trade-off between the base and new class for these methods, e.g., CoOp outperforms TCP on base classes but significantly underperforms TCP on new classes. Notably, DOR can continually reduce mis-calibration on new classes across various metrics without calibration trade-offs on base and new classes. In sum-mary, our proposed DOR can consistently boost calibration performance on new classes upon existing state-of-the-art prompt tuning methods without compromising the vanilla fine-tuning objectives. Due to space constraints, we pro-vide detailed calibration results of all datasets in Appendix F. Moreover, to illustrate how DOR affect the probability distribution, we provide a visualization in Appendix G.

**DOR effectively breaks the calibration trade-off.** To di-rectly demonstrate the role of DOR in the regularization during CLIP fine-tuning, we compare DOR and KgCoOp

Table 3. Average calibration results of ECE (%) using various regularization on 11 datasets. DOR breaks the calibration trade-off between base and novel classes.

| Method | Variant | Base | New | HM |
|---|---|---|---|---|
| CoOp | Vanilla | 3.07 | 14.49 | 8.78 |
| | +KG | 5.82 | 4.48 | 5.15 |
| | +DOR | 2.47 | 6.48 | **4.47** |
| MaPLe | Vanilla | 2.75 | 5.46 | 4.11 |
| | +KG | 4.01 | 4.29 | 4.15 |
| | +DOR | 3.06 | 4.26 | **3.66** |
| CoPrompt | Vanilla | 2.60 | 5.96 | 4.28 |
| | +KG | 4.01 | 4.99 | 4.50 |
| | +DOR | 2.98 | 5.14 | **4.06** |

Table 4. Average calibration results of ECE (%) on 11 datasets. DOR$^\dagger$ denotes that the outlier pool excludes all new classes.

| Method | Variant | Base | New | HM |
|---|---|---|---|---|
| CoOp | Vanilla | 3.07 | 14.49 | 8.78 |
| | +DOR | 2.67 | 6.49 | 4.58 |
| | +DOR$^\dagger$ | 2.82 | 6.77 | 4.80 |
| MaPLe | Vanilla | 2.75 | 5.46 | 4.11 |
| | +DOR | 2.83 | 4.44 | 3.64 |
| | +DOR$^\dagger$ | 2.89 | 4.51 | 3.70 |
| CoPrompt | Vanilla | 2.60 | 5.96 | 4.28 |
| | +DOR | 2.96 | 4.69 | 3.83 |
| | +DOR$^\dagger$ | 2.71 | 4.93 | 3.82 |

on 3 baselines (CoOp, MaPLe, and CoPrompt), and present the results in the table 3. The results demonstrate that integrating with DOR can consistently outperform those with KgCoOp on overall performance. In particular, our method achieves much better performance than KgCoOp on Base classes, while KgCoOp performs well on New classes. This is consistent with the analysis presented in Subsection 3.2: KgCoOp anchors the confidence level, leading to underconfidence on Base classes. In short, the results demonstrate the superiority of DOR in breaking the calibration trade-off between base and novel classes.

**DOR do not rely on the overlap with new classes.** As we mentioned in Section 4, DOR samples leverages text outliers for regularization. To further analyze the potential overlap with new classes used in the test time, we exclude all new classes from the outlier pool (denoted as DOR$^\dagger$). As is shown in Table 4, the average results on 11 datasets show that DOR$^\dagger$ without overlap achieves comparable performance to DOR, significantly improving the performance on all three baselines. Therefore, the effectiveness of our method does not rely on the overlap with new classes. Moreover, we present all selected outlier texts for 6 datasets, showing almost no class overlap, especially on Stanford-Cars and FGVCAircraft in Table 10. In summary, leverages general semantic information from language space rather than memorizing target classes, which ensures its fairness and generalizability.

**DOR benefits base-to-new generalization.** To further verify that our DOR is effective on new classes and non-toxic for performance on base classes, we summarize the comparison of average test accuracy in Table 6. Similar to the evaluation of calibration, a salient observation is that our proposed DOR drastically improves base-to-new generaliza-

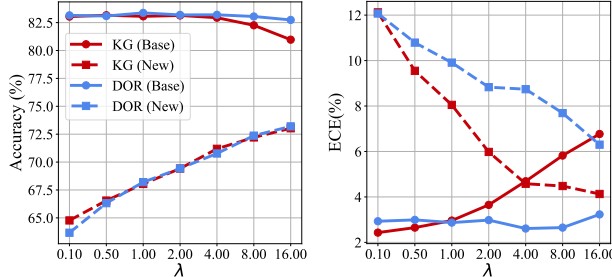

Figure 4. Parameter sensitivity between KgCoOp (KG) and DOR (w/ CoOp). Compared with KG, the accuracy and ECE of DOR are not sensitive to $\lambda$ base classes. **Left**: Accuracy. **Right**: ECE.

tion, with its accuracy consistently outperforming all existing baselines in the harmonic mean of base and new classes. Moreover, DOR almost entirely preserves model capability in the classification performance on base classes without degeneration. For instance, applying DOR in DEPT can increase accuracy on new classes from 65.04% to 71.39%, while keeping the accuracy of base classes similar to the baseline. Given that most prompt tuning methods lag behind zero-shot CLIP on the accuracy of new classes, an intuitive explanation is that DOR aligns the features of unseen classes with the zero-shot features, which can preserve the zero-shot generalization on new classes. In addition, we observe that some methods (e.g., CoCoOp and MaPLe) with DOR, resulting in improvements of 2.12% and 2.00% respectively, outperforming zero-shot accuracy on new classes. This demonstrates that DOR can significantly enhance the base-to-new generalization capacity of fine-tuned CLIP. To further demonstrate DOR can be applied to domain-specific open-vocabulary tasks, we provide more results on medical datasets in Appendix J.

**DOR is robust to covariate shifts.** To comprehensively

*Table 5.* Accuracy comparison on domain generalization datasets. DOR boosts the calibration and generalization of existing methods.

| | ECE (↓) | | | | | Accuracy (↑) | | | | |
| | Source | Target | | | | Source | Target | | | |
| | ImageNet | -V2 | -S | -A | -R | AVG | ImageNet | -V2 | -S | -A | -R | AVG |
|---|---|---|---|---|---|---|---|---|---|---|---|---|
| CLIP | 1.86 | 2.44 | 4.88 | 8.34 | 3.51 | 4.79 | 66.73 | 60.87 | 46.09 | 47.81 | 73.98 | 57.19 |
| CoOp | 1.10 | 4.19 | 8.40 | 15.34 | 0.80 | 7.18 | 71.44 | 63.55 | 45.76 | 47.81 | 73.74 | 57.72 |
| **+DOR(ours)** | 1.64 | **1.95** | **4.97** | **11.07** | 1.58 | **4.89** | 71.47 | **64.47** | **48.28** | **50.12** | **76.05** | **59.73** |
| MaPLe | 1.13 | 2.56 | 4.88 | 12.42 | 1.06 | 5.23 | 72.05 | 64.57 | 48.78 | 47.66 | 76.61 | 59.41 |
| **+DOR(ours)** | 1.46 | **1.89** | **3.96** | **11.08** | 1.37 | **4.58** | 71.93 | **64.94** | 48.77 | **48.29** | 76.20 | **59.55** |

*Table 6.* Average accuracy (%) across 11 base-to-new datasets. "Vanilla" denotes the baseline w/o DOR. DOR can improve the generalization capacity on unseen classes while maintaining the performance on base classes.

| | ZSCLIP | CoOp | | CoCoOp | | MaPLe | | KgCoOp | | DEPT | | CoPrompt | | PromptKD | |
| Class | Vanilla | Vanilla | +DOR | Vanilla | +DOR | Vanilla | +DOR | Vanilla | +DOR | Vanilla | +DOR | Vanilla | +DOR | Vanilla | +DOR |
|---|---|---|---|---|---|---|---|---|---|---|---|---|---|---|---|
| Base | 69.49 | 82.97 | **83.20** | 80.57 | 79.89 | 82.11 | 82.08 | 82.29 | 82.13 | 83.70 | **83.81** | 82.32 | 82.39 | 85.74 | 85.52 |
| New | 74.32 | 61.74 | **72.01** | 72.47 | **74.59** | 73.89 | **75.89** | 72.21 | **73.14** | 65.04 | **71.39** | 73.29 | **74.50** | 79.80 | **80.81** |
| HM | 71.90 | 72.36 | **77.61** | 76.52 | **77.24** | 78.00 | **78.98** | 77.25 | **77.64** | 74.37 | **77.60** | 77.81 | **78.44** | 82.77 | **83.17** |

verify the robustness of DOR, we further evaluate its performance in the domain generalization setting, i.e., there is a *covariate shift* between training and testing datasets. Specifically, we first fine-tune the model with all classes of ImageNet on the 16-shot setting and then evaluate it on 4 types of datasets with covariate shifts. As shown in Table 5, the calibration performance indicates DOR maintains stability in the presence of covariate shifts. Although DOR is not specially designed for the covariate shift, it outperforms the calibration baseline of CoOp and MaPLe, reducing ECE by 2.29% and 0.65% under distribution shifts respectively. Meanwhile, DOR demonstrates superior accuracy in domain generalization and successfully maintains in-distribution performance. Such regularization will not hinder the generalization capacity of fine-tuned CLIP.

**DOR is insensitive to hyperparameters.** To further illustrate the influence of hyperparameter $\lambda$, we present a sensitivity analysis. We report the average performance on 11 datasets under the base-to-new evaluation protocol. As shown in Figure 4, we can observe that DOR demonstrates robustness in model calibration as $\lambda$ in Eq.(10) varies. Although KgCoOp has better calibration on new classes as $\lambda$ increases, it sacrifices accuracy and calibration on base classes while our method does not. The results verify that DOR is an effective approach to boosting calibration performance on new classes while maintaining performance on base classes. We provide more ablations on DOR including selection strategy, similarity metric, outlier numbers, update frequency, and outliers databases in Appendix H.

# 6. Discussion

**Can the criterion of DOR be extended to visual tuning?** In this paper, we primarily focus on prompt tuning and analyze how textual divergence impacts confidence calibration. Such analysis may limit the potential scope of CLIP fine-tuning methods. To address this, we further consider a similar regularization approach based on image outliers for fine-tuning on visual representation. Specifically, we use ImageNet-1k (Deng et al., 2009) as an outlier repository and conduct experiments on four downstream datasets. To avoid potential semantic overlap between the outlier and fine-tuning data, we construct the outlier set by retaining images from 50% of the classes of ImageNet-1k, which ensures these classes differ as much as possible from those used during fine-tuning. For visual representation fine-tuning methods, we utilize CLIP-adapter (Gao et al., 2024) and visual prompt tuning (VPT) (Jia et al., 2022).

As shown in Table 7, we observe that visual-based DOR can successfully reduce the calibration error on new classes. For example, DOR outperforms VPT and CLIP-adapter baseline by reducing ECE by 4.64% and 1.82% on the DTD dataset, respectively. We provide an additional analysis of visual distribution in Appendix K. In general, visual-based DOR achieves better average calibration across various downstream datasets and leaves space for further improvement. Given that expected image outliers are not always accessible easily as text, a potential method is to generate them by diffusion (Du et al., 2024) for high-quality image outliers.

*Table 7.* Calibration results of ECE (%) on fine-tuning of visual representation. DOR-V(ision) is effective for better calibration via visual representation regularization. HM refers to harmonic mean.

| Method | Flowers Base | Flowers New | Cars Base | Cars New | DTD Base | DTD New | UCF101 Base | UCF101 New | AVG Base | AVG New | HM |
|---|---|---|---|---|---|---|---|---|---|---|---|
| VPT | 7.98 | 8.20 | 5.32 | 1.93 | 2.54 | 13.04 | 4.04 | 4.79 | 4.97 | 6.99 | 5.98 |
| **+DOR-V(ours)** | 8.19 | **7.54** | 4.90 | **1.78** | 2.68 | **8.40** | **3.73** | **4.58** | 4.88 | **5.58** | **5.23** |
| CLIP-adapter | 3.70 | 6.55 | 6.09 | 5.73 | 3.00 | 7.45 | 4.04 | 7.09 | 4.21 | 6.71 | 5.46 |
| **+DOR-V(ours)** | 4.02 | **4.86** | 7.13 | **4.85** | 3.25 | 5.63 | **1.90** | **5.23** | 4.08 | **5.14** | **4.61** |

*Table 8.* Calibration results of ECE (%) with different outliers. Oracle* denotes new classes that meet during test time.

| | w/o | Near | Far | Random | Oracle* |
|---|---|---|---|---|---|
| Base | 3.07 | **2.68** | 2.95 | 2.80 | 3.13 |
| New | 14.58 | 7.09 | 7.72 | 7.33 | **4.34** |
| HM | 8.83 | 4.89 | 5.34 | 5.07 | **3.74** |

**What makes a good regularization for CLIP fine-tuning?** In the experiments, we use the relevant but non-overlapped outlier with the fine-tuning task (referred to as near-OOD). Such selection raises the question: *why do we prefer to choose near-OOD?* To address this, we conducted an ablation on the policies of outlier selection. We considered four types of outliers: near-OOD (ours), far-OOD, random-OOD, and new classes used at test time. The far-OOD is selected by applying the opposite operation of Eq. 7. "w/o" denotes the baseline without outlier regularization. Since target new classes are unknown during fine-tuning, we view them as an oracle, which serves as a performance upper bound for base-to-new tasks. We report the average performance on the base-to-new datasets.

We present the results in Table 8. Since we primarily fine-tune the model for a specific downstream task, selecting random data or far-OOD data may not be optimal for base-to-new evaluation. Additionally, the target class is unknown during the fine-tuning phase. Therefore, we dynamically use near-OOD as regularization data, which reduces calibration errors on new classes while preserving performance on base classes. We present several word extracted by near-OOD in Appendix E. *Interestingly, the random selection can serve as a strong baseline, demonstrating the robustness of our proposed regularization item.* Moreover, while the oracle can achieve impressive performance on the calibration of new classes, the fixed number of outliers may cause the model to overfit them, resulting in a decline in generalization.

## 7. Conclusion & Limitations

In this paper, we introduce Dynamic Outlier Regularization (DOR), a simple yet effective technique that enhances the confidence calibration on both base and new classes. We show that current prompt tuning methods typically lead to a tradeoff between the base and new classes. Through the textual divergence, we provide a thorough explanation for the limitations of those tuning methods. By utilizing relevant but non-overlapped outliers, DOR regularizes the textual distribution to preserve calibration capacity in zero-shot CLIP. Our method is compatible with existing prompt-tuning methods and can be extended to improve visual fine-tuning methods, like adapters. We hope future research can extend the insight in this work to other VLMs.

**Limitations.** Similar to previous regularization methods of CLIP, DOR uses a hyperparameter $\lambda$ to control the weight of regularization, which will require extra computational costs. Moreover, our analysis is limited in the scope of CLIP, leaving other kinds of VLMs to be explored in the future.

## Impact Statement

Foundational model plays an important role in today's machine learning research. These models typically show remarkable zero-shot generalization capabilities and achieve better performance on downstream tasks via fine-tuning. Unfortunately, it often comes at the cost of generalization, which posing a significant challenge in real-world applications. In this paper, we investigate the confidence calibration of vision-language models (e.g., CLIP) after fine-tuning. Our goal is to mitigate overconfidence while preserving the models' adaptability to diverse tasks. We hope our findings can extend to larger language models or more vision-language architectures, aiming to enhance their robustness after post-training in real-world scenarios.

## Acknowledgment

Shuoyuan Wang and Hongxin Wei are supported by the Shenzhen Fundamental Research Program (Grant No. JCYJ20230807091809020). We gratefully acknowledge the support of the Center for Computational Science and Engineering at the Southern University of Science and Technology for our research.

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

# A. Related work

**Vision-language models.** Large pre-trained large vision-language models (Jia et al., 2021; Radford et al., 2021) have been verified to effectively comprehend visual concepts using language supervision and apply them in downstream tasks (e.g. image classification (Radford et al., 2021; Zhou et al., 2022b; Lu et al., 2022; Naeem et al., 2023), knowledge-augmented retrieval(Ming & Li, 2024) and visual question answering (Parelli et al., 2023)) in a zero-shot manner. Despite VLM's effectiveness in generalizing new visual concepts, the performance of zero-shot CLIP still lags behind the fine-tuned performance on specific downstream tasks (Zhang et al., 2024a). To further boost the downstream adaptation of pre-trained VLMs, many parameter-efficient tuning methods like vanilla prompt tuning (Zhou et al., 2022b;a; Khattak et al., 2023a) and adapter tuning (Gao et al., 2024; Zhang et al., 2022) have been proposed for high efficiency. Moreover, many regularization-based tuning methods have been proposed to preserve the generalization performance on unseen classes (Yao et al., 2023; 2024; Zhu et al., 2023; Roy & Etemad, 2023; Khattak et al., 2023b). Despite the greater success of CLIP fine-tuning, the effectiveness of safety-related evaluation like confidence calibration has largely been overlooked, which is essential for real-world deployment.

**Confidence calibration.** Confidence calibration has been widely studied to ensure that the confidence levels output by models accurately reflect their empirical accuracy. To achieve this, the state-of-the-art calibration methods can be categorized into regularization methods and post-hoc methods. For regularization methods, they either explicitly or implicitly regularize modern neural networks to have better calibration. Although regularization methods may not designed for calibration, they generally have better calibration performance including L2 regularization (Guo et al., 2017), Entropy regularization (Pereyra et al., 2017), focal loss (Mukhoti et al., 2020), etc. On the other hand, post-hoc methods fix the output probability after the training phase. For post-hoc methods, the most representative and simple method is temperature scaling (Guo et al., 2017), which learns a single scalar for rescaling the softmax logit. ATS (Joy et al., 2023) modifies the predicted confidence by per-data-point adaptive temperature. Another type of post-hoc calibration is binning-based calibration (Zadrozny & Elkan, 2001; 2002). For instance, Mix-n-Match (Zhang et al., 2020) leverages ensemble and composition techniques to achieve data efficiency and maintain accuracy in confidence estimates. Recently, several works have explored the calibration in CLIP. (Murugesan et al., 2025) investigate the calibration of CLIP under covariate shift. Given that existing post-hoc calibration on base classes can not transfer to new classes, DAC (Wang et al., 2024) fixes the logit scale of prediction via textual deviation-informed score in a post-hoc manner. Different from them, we address the calibration issue during the fine-tuning phase. In this work, we introduce a regularization based on dynamic outliers. We demonstrate that DOR boosts the calibration performance of many existing state-of-the-art prompt tuning methods of CLIP without affecting the vanilla fine-tuning objective.

**Outlier regularization in trustworthy machine learning.** The outlier plays an important role in trustworthy machine learning research including out-of-distribution (OOD) detection, noisy label learning, adversarial attack, long-tailed datasets re-balancing, etc. In OOD detection, outliers are typically used to simulate the distribution of OOD data, thereby increasing the distinction between ID data and OOD data (Hendrycks et al., 2019; Liu et al., 2020; Ming et al., 2022; Jiang et al., 2024). Recently, Dream-OOD generate the expected OOD data by diffusion (Du et al., 2024). In noisy label learning, ODNL (Wei et al., 2021) leverages outliers as dynamical noisy labels to improve model robustness against noisy labels. To address the extreme class imbalance, Open-sampling (Wei et al., 2022) re-balance class priors via open-set noisy labels. OAT (Lee et al., 2021) leverages outlier data to improve model generalization in adversarial robustness, which regularizes the softmax probabilities to be a uniform distribution for outliers. In this paper, we utilize textual outliers to control the divergence of unseen textual distribution, and further improve the calibration of fine-tuned CLIP.

# B. Additional analysis of the motivation

## B.1. Detailed Results of textual divergence

In section 3.2, we mainly derive the reason for the miscalibration issue from CoOp and KgCoOp. specifically, We use FD score to measure the diversity of textual representation and evaluate the prompt tuning methods on the value of output confidence and logit gap. To comprehensively verify our motivation, we further compare more prompt tuning methods including CoCoOp, MaPLe, PromptSRC and CoPrompt.

As is in Figure 5a, we can observe a similar phenomenon as we discussed in section 3.2. We find that fine-tuning can significantly increase the FD score of textual features compared to the zero-shot CLIP. Notably, this observation can extend

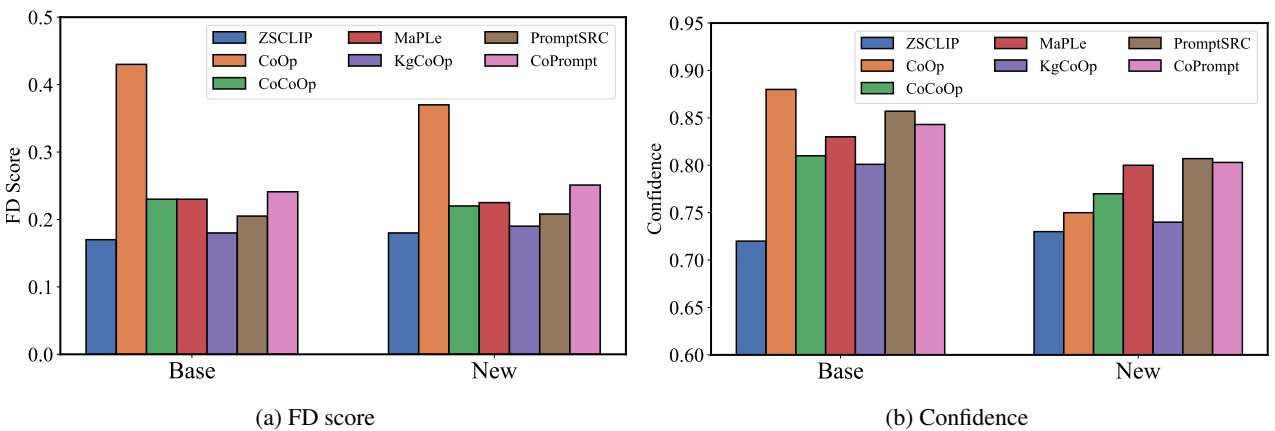

(a) FD score                                   (b) Confidence

*Figure 5.* Comparison between zero-shot CLIP and different prompt tuning methods on UCF101 dataset. Fine-tuned CLIP tends to have higher confidence and FD score on both base and new classes.

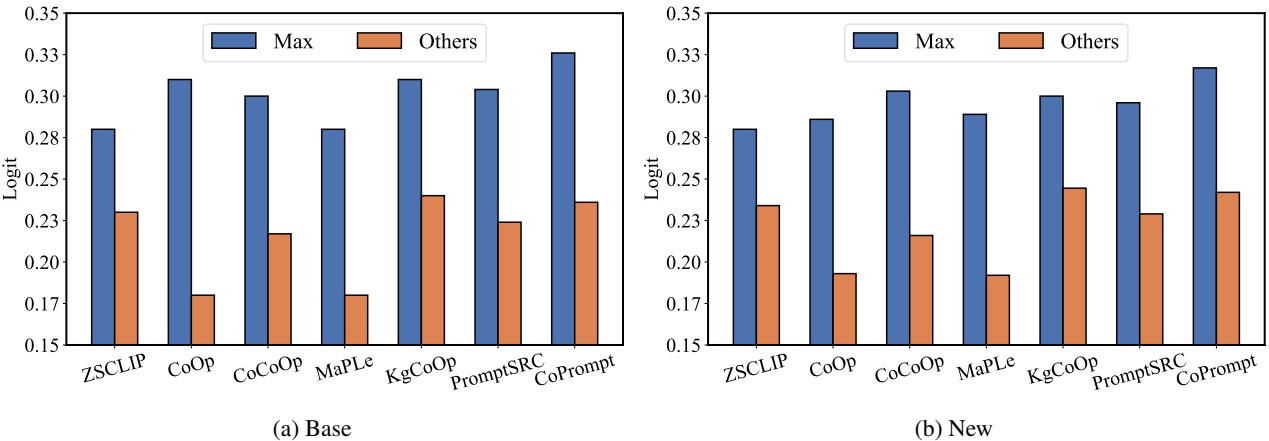

(a) Base                                       (b) New

*Figure 6.* Comparison between the maximum logit and the average of other logits on DTD dataset. The logit gap become wider in fine-tuned CLIP.

to new classes, even if they are not explicitly optimized during the fine-tuning phase. Consequently, the gap between the maximum logit and the others widens after fine-tuning in Figure 6. Therefore, the tuned CLIP with CE loss will make softmax predictions with high confidence, which can generate higher average confidence (See Figure 5b). Such an increase in confidence aligns with the improved accuracy on base classes. However, it is not consistent with the nearly unchanged accuracy on new classes, which leads to overconfidence.

## B.2. Why using outliers for the regularization?

To further demonstrate the superiority of outliers in the regularization for CLIP fine-tuning, we provide additional analysis in this section. Specifically, we first analyze the calibration issue from the perspective of gradient conflicts (Shi et al., 2023) and present empirical evidence to understand previous regularization terms hinder the calibration of base classes. We can decouple CLIP's optimization objective as

$$\mathcal{L}_{\text{clip}} = \mathcal{L}_{\text{ce}} + \lambda \cdot \mathcal{L}_{\text{reg}},$$

where $\mathcal{L}_{\text{ce}}$ is the cross-entropy loss for classification, and $\mathcal{L}_{\text{reg}}$ denotes the regularization term. Previously proposed regularization terms include $\mathcal{L}_{\text{kg}}$ (KgCoOp), $\mathcal{L}_{\text{distill}}$ (CoPrompt), and $\mathcal{L}_{\text{scl}}$ (PromptSRC). For reasonable comparison, we set hyperparameters to be the same including optimizer, learning rate, etc. We calculate the cosine similarity of the prompt

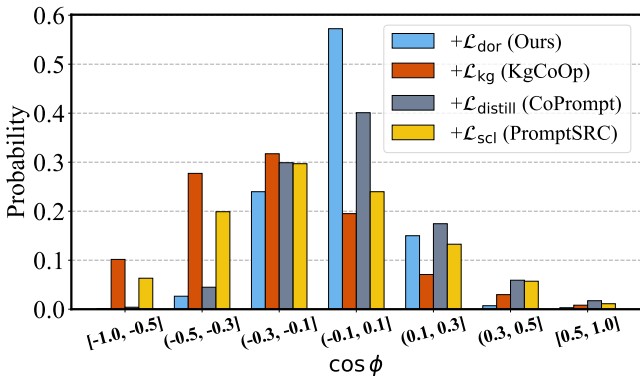
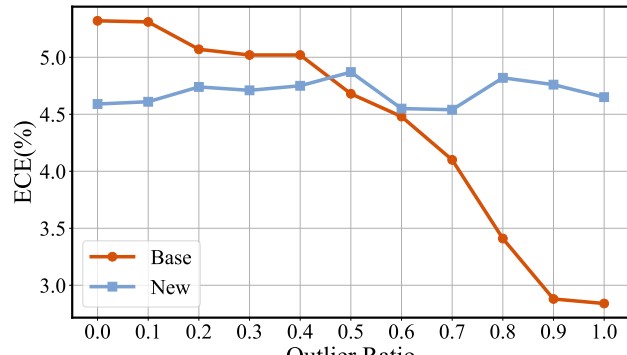

*Figure 7.* The distributions of gradient conflicts (in terms of $\cos\phi$). DOR shows less gradient conflict compared with recent prompt tuning methods.

*Figure 8.* ECE comparison as the proportion of outliers changed. Outliers can break the calibration trade-off on base and new classes.

gradients between $\mathcal{L}_{\mathrm{reg}}$ and $\mathcal{L}_{\mathrm{ce}}$ to reflect the degree of gradient conflicts.

**Regularization from base classes hinders original fine-tuning objective.** As shown in Figure 7, the gradient conflict distributions for KgCoOp, CoPrompt, and PromptSRC are predominantly within the range of $[-1, 0]$, which indicates a conflict with the original learning objective. Considering that CoOp with vanilla $\mathcal{L}_{\mathrm{ce}}$ is an efficient calibrator for base classes, we can infer that these regularization terms may hinder the calibration performance for base classes. As an alternative, our proposed DOR leverages outliers to construct the regularization term. We observe that the gradient conflicts for DOR are primarily concentrated within the range $(-0.1, 0.1]$. Compared to previous regularization terms, it shows significantly fewer conflicts in the $[-1, 0]$ range. The phenomenon supports our claim that outliers can be used in regularization without interfering with the original fine-tuning objective.

**Outlier-based regularization can break the calibration trade-off.** To further illustrate the actual performance of outliers in calibration, we conducted an analysis based on KgCoOp. Since KgCoOp uses a fixed number of base classes as the regularization term, we progressively replaced these texts with textual outliers at varying proportions $[0.1, 0.2...1.0]$. As shown in Figure 8, as the proportion of outliers increases, the calibration of base classes improves while the performance on new classes remains largely unaffected. Such observation strongly supports our claim that outlier can effectively mitigate the miscalibration issue on new classes while maintaining the calibration performance on the base classes.

## C. Theoretical justification

To help readers understand the insights, we formalize our observations of CLIP fine-tuning that the textual divergence is a significant factor for confidence estimation in this part. As the image feature remains unchanged in fine-tuning, the textual divergence can be translated to the variance of logits, which is computed by the similarity between the image feature and different text features. In the following, we formally show the relationship between the logit variance and the confidence.

For simplicity, we consider a binary classification problem. Let $\{z_i\}_{i=1}^{n}$ be a set of logit vectors (i.e., model outputs), where each vector $z_i = [z_1, z_2]^{\mathrm{T}}$ consists of two logits. We assume that the logit $z$ is an independent random variable drawn from a normal distribution $\mathcal{N}(\mu, \sigma^2)$. The confidence (i.e., maximum softmax probability) $p_i$ is given by the softmax (or sigmoid) function defined as in Eq.(2). We have the following proposition.

**Proposition C.1.** *Let $\mathbb{E}[p_\sigma]$ denote the expected value of the maximum probability $p_i$ when the logits are distributed as $\mathcal{N}(\mu, \sigma^2)$. Then, for any $\sigma_1, \sigma_2 > 0$ and $\mu$, we have $\mathbb{E}[p_{\sigma_2}] > \mathbb{E}[p_{\sigma_1}]$, if $\sigma_2 > \sigma_1$.*

This suggests that the high divergence in the logit distribution tends to generate larger predicted confidence, which is induced by the textual divergence using CE loss. In Section 5, we empirically verify that our proposed method can preserve the textual divergence on the new classes, thereby improving the calibration performance.

*Proof.* We first remove the influence of $\mu$. For any constant $c$, we have:

$$p_i = \frac{e^{z_i + c}}{\sum_{j=1}^{N} e^{z_j + c}} = \frac{e^{z_i}}{\sum_{j=1}^{N} e^{z_j}}.$$

Thus, the mean value $\mu$ of the logits does not affect the softmax output. Therefore, without loss of generality, we can assume that $\mu = 0$ in our proof. For binary classification, we have:

$$p_1 = \frac{e^{z_1}}{e^{z_1} + e^{z_2}} = \frac{1}{1 + e^{-(z_1 - z_2)}} = \text{sigmoid}\,(z_1 - z_2)\,, p_2 = \text{sigmoid}\,(z_2 - z_1)\,.$$

Hence, the maximum softmax probability is:

$$p_{\max} = \max(p_1, p_2) = \text{sigmoid}(|z_1 - z_2|).$$

Since $z_1 - z_2 \sim \mathcal{N}(0, 2\sigma^2)$, the absolute difference $Z = |z_1 - z_2|$ follows a folded normal distribution with the probability density function:

$$f_Z(z) = \frac{1}{\sqrt{\pi\sigma^2}} e^{-\frac{z^2}{4\sigma^2}}, \quad z \geq 0.$$

Thus, the expected value of the maximum softmax probability is:

$$E[p_{\max}] = E[\text{sigmoid}(Z)] = \int_0^\infty \text{sigmoid}(z) \cdot f_Z(z)dz = \int_0^\infty \frac{1}{1 + e^{-z}} \cdot \frac{1}{\sqrt{\pi\sigma^2}} e^{-\frac{z^2}{4\sigma^2}} dz.$$

For simplicity, we perform the substitution $u = \frac{z}{\sigma\sqrt{2}}$. The integral can be simplified to:

$$E[p_{\max}] = \int_0^\infty \frac{1}{1 + e^{-\sigma\sqrt{2}u}} \cdot \frac{\sqrt{2}}{\sqrt{\pi}} e^{-\frac{u^2}{2}} du.$$

Using Leibniz's rule, we can differentiate with respect to $\sigma$ under the integral sign:

$$\frac{dE[p_{\max}]}{d\sigma} = \int_0^\infty \frac{\partial}{\partial\sigma} \left( \frac{1}{1 + e^{-\sigma\sqrt{2}u}} \right) \cdot \frac{\sqrt{2}}{\sqrt{\pi}} e^{-\frac{u^2}{2}} du = \int_0^\infty \frac{e^{-\sigma\sqrt{2}u} \cdot 2u}{\left(1 + e^{-\sigma\sqrt{2}u}\right)^2} \cdot \frac{1}{\sqrt{\pi}} e^{-\frac{u^2}{2}} du \geq 0.$$

Hence, the expected maximum probability $E[p_{\max}]$ increase alone with $\sigma$. Then, We have $\mathbb{E}[p_{\sigma_2}] > \mathbb{E}[p_{\sigma_1}]$, if $\sigma_2 > \sigma_1$. The proposition is proven. $\qquad\square$

## D. Implementation details

*Table 9.* Hyperparameters for VLM tuning methods. "BS" denotes the batch size. "LR" denotes the learning rate. "CTX" is the context length of the learnable prompt.

|  | CoOp | CoCoOp | DEPT | KgCoOp | MaPLe | TCP | CLIP-Adapter | VPT |
|---|---|---|---|---|---|---|---|---|
| Epochs | 200 | 10 | 200 | 200 | 5 | 50 | 200 | 5 |
| BS | 32 | 1 | 32 | 32 | 4 | 32 | 32 | 4 |
| LR | 0.002 | 0.002 | 0.002 | 0.002 | 0.0026 | 0.002 | 0.002 | 0.0025 |
| CTX | 16 | 4 | 16 | 16 | 2 | 4 | - | 8 |

Our implementations are based on the open-source repository of DAC (Wang et al., 2024). Generally, We use CLIP ( ViT-B/16) (Radford et al., 2021) as the pre-trained VLM throughout our experiments and report results averaged over 3 runs. We fine-tune the model with 16 samples per class in a few-shot setting (Zhou et al., 2022a). Following the corresponding

*Table 10.* Outlier selection from WordNet based on zero-shot CLIP.

| Dataset | Base class | Selected outlier |
|---|---|---|
| Flowers102 | ['pink primrose', 'hard-leaved pocket orchid', 'sweet pea', 'english marigold', 'tiger lily', 'moon orchid', 'bird of paradise', 'monkshood', 'globe thistle', 'snapdragon', "colt's foot", 'king protea', 'spear thistle', 'yellow iris', 'globe-flower', 'purple coneflower', 'peruvian lily', 'balloon flower'] | ['May_lily', 'flowering_plant', 'dayflower', 'coast_lily', 'flower', 'lily', 'orchid', 'African_lily', 'non-flowering_plant', 'plant', 'sego_lily', 'plant_material', 'flower-of-an-hour', 'flora', 'liliaceous_plant', 'Liliaceae', 'apetalous_flower', 'tongueflower', 'herbaceous_plant', 'daisybush'] |
| OxfordPets | ['abyssinian', 'american_bulldog', 'american_pit_bull_terrier', 'basset_hound', 'beagle', 'bengal', 'birman', 'bombay', 'boxer', 'british_shorthair', 'chihuahua', 'egyptian_mau', 'english_cocker_spaniel', 'english_setter', 'german_shorthaired'] | ['spaniel', 'bulldog', 'dog', 'dog_do', 'doggie', 'doggy', 'domestic_dog', 'canine', 'pug-dog', 'pooch', 'Japanese_spaniel', 'Labrador_retriever', 'sausage_dog', 'Labrador', 'Little_Dog', 'housedog', 'CAT', 'retriever', 'French_bulldog', 'bird_dog'] |
| StanfordCars | [ '2012 Acura RL Sedan', '2012 Acura TL Sedan', '2008 Acura TL Type-S', '2012 Acura TSX Sedan', '2001 Acura Integra Type R', '2012 Acura ZDX Hatchback'] | ['estate_car', 'automotive_vehicle', 'sedan', 'used-car', 'tesla', 'Tesla', 'pickup_truck', 'car', 'SUV', 'hatchback', 'subcompact_car', 'patrol_car', 'station_wagon', 'sports_car', 'sport_utility_vehicle', 'passenger_vehicle', 'secondhand_car', 'vehicle', 'sport_car', 'touring_car'] |
| FGVCAircraft | ['707-320', '727-200', '737-200', '737-300', '737-400', '737-500', '747-200', '747-300', '747-400', '757-200', A340-600', 'A380', 'ATR-72', 'BAE 146-200', 'BAE 146-300', 'BAE-125', 'Beechcraft 'Boeing 717', 'C-130', 'C-47', 'CRJ-200', 'CRJ-700', 'CRJ-900', 'Cessna 172', 'Cessna 208', 'Cessna 525'] | ['airliner', 'widebody_aircraft', 'aircraft', 'airbus', 'jetliner', 'wide-body_aircraft', 'jumbojet', 'air_transport', 'narrow-body_aircraft', 'multiengine_airplane', 'airline', 'attack_aircraft', 'reconnaissance_plane', 'air_transportation', 'military_plane', 'aeroplane', 'plane', 'multiengine_plane', ] |
| Food101 | ['apple_pie', 'baby_back_ribs', 'baklava', 'beef_carpaccio', 'beef_tartare', 'beet_salad', 'beignets', 'bibimbap', 'bread_pudding', 'breakfast_burrito', 'bruschetta', 'caesar_salad', 'cannoli', 'caprese_salad', 'carrot_cake', 'ceviche', 'cheese_plate', 'cheesecake', 'chicken_curry', 'chicken_quesadilla', | ['entree', 'pastry', 'bread', 'breakfast_food', 'food', 'salad', 'burger', 'Burger', 'dessert', 'soup', 'French_pastry', 'sandwich', 'steak', 'meat', 'pizza', 'cuisine', 'pie', 'PIE', 'French_bread', 'dish'] |
| UCF101 | ['Apply_Eye_Makeup', 'Apply_Lipstick', 'Archery', 'Baby_Crawling', 'Balance_Beam', 'Band_Marching', 'Baseball_Pitch', 'Basketball', 'Basketball_Dunk', 'Bench_Press', 'Biking'] | ['weightlifting', 'athletics', 'sports_implement', 'hitting', 'physical_exercise', 'near_thing', 'athletic_competition', 'phot', 'depicting', 'fitness', 'athletic_game', 'physical_fitness', 'batting', 'goal', 'musical_style', 'photography', 'going'] |

official implementation, We list the general hyperparameters in Table 9. For hyperparameter $\lambda$ in DOR, we set $\lambda = 8.0$ for CoOp, $\lambda = 4.0$ for MaPLe and 2.0 for other fine-tuning methods. We set the number of selected dynamic outlier repository to 5000. The number of outliers in each batch is the same as the base classes. Here, we adopt these VLM tuning methods from the corresponding official implementation and briefly introduce the corresponding hyperparameters of them. All the methods are adopted from their official implementation. For CoOp and CoCoOp, they do not contain other hyperparameters. For KgCoOp, we set $\lambda = 8.0$. For MaPLe, we set prompt depth $J$ to 0 and the language and vision prompt lengths to 2. For DePT, the learning rate for updating the parameters in the devised CAT head is set to $6.5 \times \delta$. where $\delta$ is the adopted learning rate of CoOp. Moreover, the weight in the linear probe is set to 0.7. For TCP, the weight for prompt fusion is 1.0, and the loss weight is the same as KgCoOp. For CLIP-adapter, we set $\alpha$ to 0.6, which is a trade-off hyperparameter between fine-tuned and zero-shot visual representation. Finally, following MaPLe, we set the context length to 8.0 and prompt depth to 12 for VPT.

## E. A close look at selected outliers

In this section, we present the detailed results for the dynamic outlier selection. Specifically, we select nouns from WordNet that do not overlap but share higher-level concept relations with the base classes seen in the fine-tuning task. We use the textual encoder of zero-shot CLIP as the modality encoder.

As shown in Table 10, we can conclude that the selected outlier meets our requirement. For example, if our base class contains certain aircraft models such as 'A340-600', 'A380', and 'ATR-72', the outliers we selected include words like 'air_transport', 'air_transportation', and 'military_plane'. These nouns are highly relevant to the downstream task but do not overlap with the base class. For further influence, we show that using outliers that are relevant but do not overlap with our fine-tuning task are helpful in reducing calibration error while preserving performance in the base classes. To verify it, we

empirically demonstrate that using base classes as the regularization may sacrifice its accuracy and calibration in Figure 4. Moreover, As shown in Table 8, dynamic text is better than fixed text since the fixed number of text may cause the model to overfit them.

## F. The detailed experimental results

In this section, We present the detailed results of Expected Calibration Error (ECE) to verify the effectiveness of our proposed DOR in Table 11.

*Table 11.* ECE (%) comparison of existing prompt tuning in the base-to-new generalization.

| Methods | Caltech101 | Pets | Cars | Flowers | Food101 | FGVC | SUN397 | DTD | EuroSAT | UCF101 | ImageNet | AVG |
|---|---|---|---|---|---|---|---|---|---|---|---|---|
| ZeroshotCLIP | 6.49 | 2.25 | 3.74 | 3.11 | 1.57 | 3.03 | 1.59 | 4.53 | 8.35 | 3.24 | 1.51 | 3.58 |
| CoCoOp | 1.45 | 2.32 | 6.61 | 7.67 | 1.10 | 3.41 | 1.66 | 2.61 | 8.06 | 2.08 | 2.66 | 3.60 |
| CoCoOp+DOR | 2.20 | 2.97 | 6.85 | 8.49 | 0.97 | 3.33 | 3.19 | 2.86 | 10.12 | 2.96 | 2.52 | 4.22 |
| CoOp | 0.95 | 0.96 | 2.59 | 2.38 | 2.79 | 5.84 | 4.57 | 6.73 | 1.50 | 4.04 | 1.38 | 3.07 |
| CoOp+DOR | 1.59 | 1.78 | 4.20 | 4.90 | 0.63 | 3.35 | 0.87 | 5.81 | 2.66 | 1.61 | 1.96 | 2.67 |
| DEPT | 2.22 | 6.83 | 12.08 | 7.75 | 5.41 | 5.23 | 2.90 | 3.29 | 8.26 | 3.32 | 9.15 | 6.04 |
| DEPT+DOR | 2.95 | 8.32 | 13.37 | 10.26 | 7.04 | 6.51 | 5.55 | 4.78 | 9.50 | 5.08 | 10.96 | 7.67 |
| KgCoOp | 2.30 | 2.95 | 11.42 | 10.05 | 1.35 | 5.40 | 4.69 | 8.02 | 10.97 | 4.18 | 2.65 | 5.82 |
| KgCoOp+DOR | 2.48 | 2.96 | 11.02 | 10.19 | 1.39 | 7.64 | 4.84 | 8.34 | 11.03 | 4.32 | 2.57 | 6.07 |
| MaPLe | 1.21 | 2.09 | 5.81 | 4.23 | 0.82 | 3.46 | 1.04 | 4.34 | 3.53 | 1.77 | 1.95 | 2.75 |
| MaPLe+DOR | 1.84 | 2.05 | 6.49 | 4.47 | 0.86 | 2.40 | 2.28 | 2.32 | 4.44 | 3.08 | 2.02 | 2.93 |
| TCP | 1.99 | 2.44 | 8.93 | 6.83 | 1.56 | 5.28 | 2.64 | 6.83 | 9.58 | 3.58 | 2.12 | 4.71 |
| TCP+DOR | 2.03 | 2.46 | 9.34 | 7.10 | 1.63 | 5.28 | 2.90 | 6.52 | 9.84 | 3.48 | 2.09 | 4.79 |
| PromptSRC | 2.41 | 2.37 | 8.14 | 4.62 | 0.89 | 4.29 | 2.08 | 2.87 | 9.15 | 2.43 | 2.01 | 3.75 |
| PromptSRC+DOR | 2.32 | 2.56 | 8.19 | 4.92 | 0.95 | 4.12 | 2.16 | 3.53 | 9.20 | 2.59 | 2.13 | 3.88 |
| CoPrompt | 1.56 | 2.81 | 3.91 | 4.92 | 0.99 | 2.47 | 0.90 | 2.78 | 4.05 | 2.10 | 1.68 | 2.56 |
| CoPrompt+DOR | 1.96 | 2.86 | 4.25 | 6.24 | 1.02 | 2.50 | 1.53 | 2.97 | 5.63 | 1.74 | 1.91 | 2.96 |

(a) Base

| Methods | Caltech101 | Pets | Cars | Flowers | Food101 | FGVC | SUN397 | DTD | EuroSAT | UCF101 | ImageNet | AVG |
|---|---|---|---|---|---|---|---|---|---|---|---|---|
| ZeroshotCLIP | 1.60 | 3.42 | 3.31 | 4.91 | 1.83 | 6.55 | 3.48 | 8.89 | 9.12 | 5.52 | 2.09 | 4.61 |
| CoCoOp | 3.94 | 2.35 | 2.26 | 11.33 | 1.63 | 12.51 | 2.03 | 16.40 | 9.17 | 4.39 | 1.57 | 6.14 |
| CoCoOp+DOR | 1.30 | 2.98 | 3.03 | 6.77 | 1.82 | 7.67 | 1.12 | 6.00 | 8.26 | 3.65 | 1.67 | 4.02 |
| CoOp | 4.11 | 1.57 | 11.81 | 19.84 | 4.42 | 32.12 | 15.98 | 26.49 | 15.50 | 18.09 | 10.40 | 14.58 |
| CoOp+DOR | 1.42 | 2.94 | 8.01 | 7.34 | 1.22 | 21.21 | 2.31 | 11.34 | 8.67 | 5.07 | 1.89 | 6.49 |
| DEPT | 4.23 | 2.71 | 11.15 | 18.32 | 2.71 | 34.21 | 15.15 | 24.30 | 18.90 | 18.94 | 9.73 | 14.58 |
| DEPT+DOR | 2.51 | 2.64 | 7.53 | 6.58 | 0.74 | 22.62 | 5.01 | 17.10 | 7.34 | 7.65 | 2.77 | 7.50 |
| KgCoOp | 2.02 | 3.15 | 3.35 | 5.92 | 1.87 | 12.76 | 1.51 | 7.41 | 6.56 | 2.95 | 1.74 | 4.48 |
| KgCoOp+DOR | 1.43 | 3.04 | 3.46 | 6.68 | 1.83 | 9.63 | 2.33 | 5.78 | 5.29 | 2.52 | 1.86 | 3.99 |
| MaPLe | 2.66 | 2.35 | 2.95 | 10.32 | 1.16 | 10.72 | 2.42 | 15.54 | 6.06 | 3.65 | 2.27 | 5.46 |
| MaPLe+DOR | 1.71 | 2.50 | 2.42 | 10.27 | 1.51 | 10.56 | 0.90 | 10.64 | 7.15 | 2.52 | 1.68 | 4.71 |
| TCP | 1.15 | 2.94 | 2.46 | 5.14 | 2.34 | 8.07 | 1.98 | 4.91 | 8.36 | 5.78 | 1.59 | 4.07 |
| TCP+DOR | 1.21 | 3.03 | 2.43 | 4.26 | 2.23 | 7.51 | 2.56 | 4.72 | 6.21 | 5.91 | 1.70 | 3.80 |
| PromptSRC | 1.55 | 3.07 | 2.02 | 5.53 | 1.66 | 11.31 | 0.66 | 6.42 | 8.53 | 3.24 | 1.71 | 4.15 |
| PromptSRC+DOR | 1.64 | 2.82 | 1.84 | 5.58 | 1.49 | 9.58 | 0.74 | 5.72 | 7.56 | 3.04 | 1.82 | 3.80 |
| CoPrompt | 1.69 | 2.41 | 5.59 | 10.19 | 1.67 | 11.54 | 2.28 | 8.64 | 16.18 | 2.60 | 2.79 | 5.96 |
| CoPrompt+DOR | 1.43 | 3.13 | 5.50 | 8.48 | 1.70 | 13.16 | 1.17 | 5.68 | 6.28 | 2.66 | 2.40 | 4.69 |

(b) New

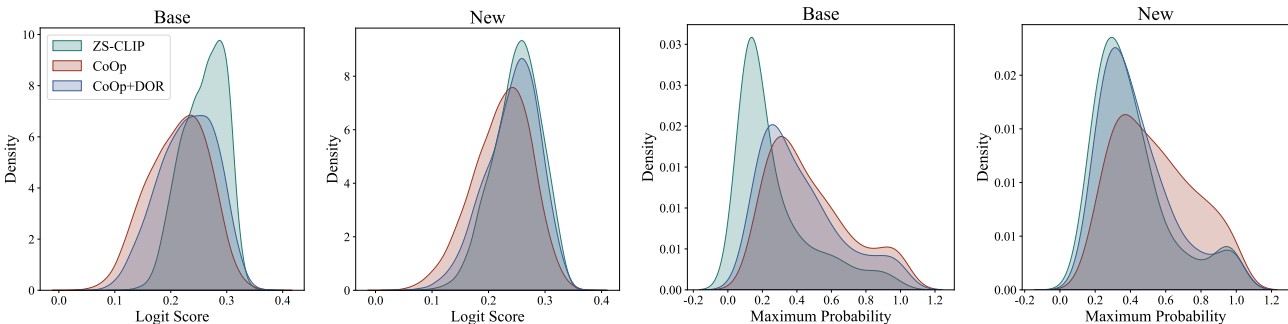

*Figure 9.* Distribution visualization of logit and maximum softmax probability on FGVCAircraft dataset. Our method generates logit and probability distributions that closely resemble those of CoOp for the base class and are similar to zero-shot CLIP for the new class.

*Table 12.* Average calibration performance (ECE%) across 11 datasets using different similarity metrics. Calibration error is given by $\times 10^{-2}$. "HM" denotes the harmonic mean.

| Metric | CoOp | | | MaPLe | | | CoPrompt | | |
|---|---|---|---|---|---|---|---|---|---|
| | Base | New | HM | Base | New | HM | Base | New | HM |
| w/o DOR | 3.07 | 14.58 | 8.83 | **2.75** | 5.46 | 4.11 | **2.56** | 5.96 | 4.26 |
| Cosine | **2.67** | 6.49 | 4.58 | 2.93 | 4.71 | 3.82 | 2.96 | 4.96 | 3.96 |
| Euclidean | 2.92 | **4.74** | **3.83** | 2.83 | **4.63** | **3.73** | 2.92 | 4.78 | **3.85** |
| Mahalanobis | 2.83 | 6.73 | 4.78 | 3.05 | 4.86 | 3.96 | 2.96 | **4.76** | 3.86 |

# G. How does DOR affect the distribution of logit and probability?

To further illustrate the influence of DOR on confidence calibration, we visualize and compare the distribution of output logit and softmax confidence score for base and new classes in Figure 9. We compare zero-shot CLIP and CoOp w/ or w/o DOR on FGVCAircraft dataset. The results verify that DOR modifies the logit distribution and confidence level. We can observe that if the model tuning With CoOp+DOR, the logit distribution of base class approximate CoOp, and the new class is similar to zero-shot CLIP, respectively. A similar phenomenon is observed in the softmax probability distribution. The results meet our vision mentioned in Section 3.1, DOR can leverage the advantages of both models, which ensure the confidence calibration on both base and new classes after fine-tuning.

# H. The ablations of DOR

### H.1. The similarity metric in outlier selection

In the outlier selection, we use cosine similarity as the distance metric for selecting textual outliers. To assess the metric sensitivity of our proposed DOR, we conduct an ablation and use three metrics including Cosine similarity, Euclidean distance (L2), and Mahalanobis distance. We report the average calibration performance on the base-to-new datasets across various prompt tuning methods, including CoOp, MaPLe, and CoPrompt.

We present the results in Table 12. We find that our DOR framework is not very sensitive to the choice of similarity metric and consistently reduces the calibration error. Surprisingly, Euclidean distance can outperform Cosine similarity across different prompt tuning methods and achieve better harmonic mean (HM). For example, using Euclidean distance with CoOp yielded an HM of 3.83%, compared to 4.58% with Cosine similarity. Despite this, we use Cosine similarity as the default distance metric throughout this work since it is widely used in the feature selection of vision-language models (Yi et al., 2024; Mayilvahanan et al., 2024; Wang et al., 2024).

### H.2. The size of selected outlier set

We evaluate how the number of selected outliers in DOR affects the calibration performance. Specifically, we present this ablation with average ECE on the base-to-new datasets with three prompt tuning methods including CoOp, MaPLe, and

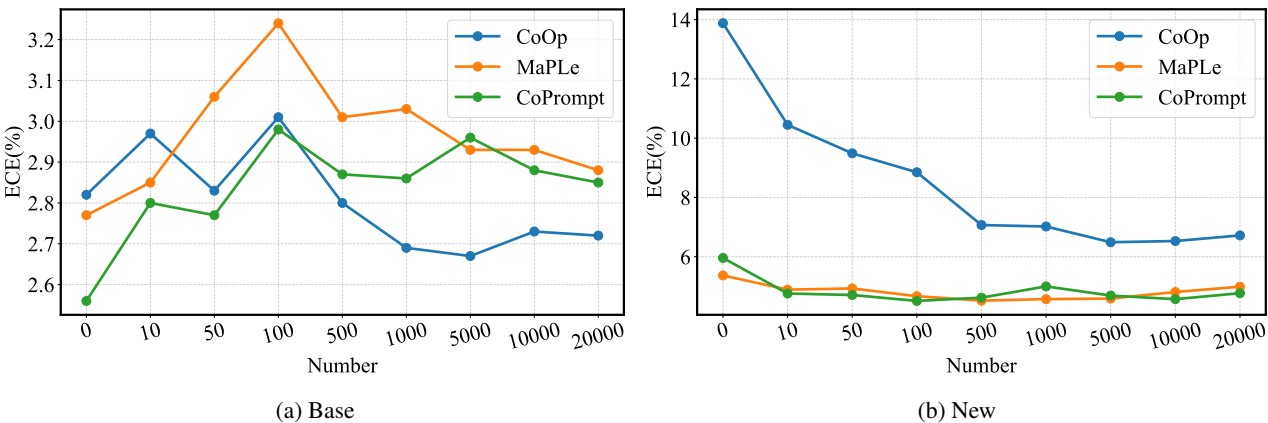

(a) Base                (b) New

*Figure 10.* The ablation on the number of selected outliers. ECE (%) is calculated from the average performance on base-to-new datasets.

*Table 13.* Average ECE (%) across 11 datasets with different frequencies of outlier update in the batch. We use "1" denotes that the outliers are updated in every iteration.

|      | CoOp |      |      |      | CoPrompt |      |      |      |
| ---- | ---- | ---- | ---- | ---- | ---- | ---- | ---- | ---- |
|      | 1    | 10   | 100  | 1000 | 1    | 10   | 100  | 1000 |
| Base | 2.67 | 2.76 | 2.71 | 2.65 | 2.96 | 3.00 | 2.80 | 2.86 |
| New  | 6.49 | 6.72 | 7.00 | 7.43 | 4.69 | 4.68 | 4.85 | 5.03 |
| HM   | 4.58 | 4.74 | 4.86 | 5.04 | 3.83 | 3.84 | 3.83 | 3.95 |

CoPrompt. We vary the number of outliers $k = \{10, 50, 100, \ldots, 20000\}$.

As shown in Figure 10, increasing the number of selected outliers leads to an evident reduction in ECE on the new classes. The performance starts to reach a point of saturation with more outliers. Notably, even setting $k = 10$ yields significant calibration improvements on the new classes. For the base classes, the outliers may be fixed in the batch during the fine-tuning due to the small number, leading to poor performance due to overfitting. We can observe a similar phenomenon in Table 1 (KgCoOp) or Table 8. Hence, we suggest to set a moderate number (e.g., 5000). Furthermore, when the number of outliers is sufficiently large, DOR becomes less sensitive to the exact value of numbers, demonstrating its robustness.

### H.3. The frequency of outliers

In DOR, we randomly sample a batch of textual outliers from the selected textual outlier set, in each iteration. Therefore, the textual outliers used in each iteration can be different, which establishes a dynamic regularization. To evaluate the impact of outlier update frequency on performance, we conduct an ablation study by varying the frequency at which outliers are sampled for the batch with the update intervals in the range $[1, 10, 100, 1000]$. We present this ablation with average ECE on the base-to-new datasets with two prompt tuning methods including CoOp and CoPrompt.

As shown in Table 13, low update frequencies (or large update intervals) are observed to negatively impact the calibration performance of DOR. The phenomenon suggests that infrequent updates may allow the model to overfit to noise and reduce its calibration performance. In short, The experimental results highlight the benefits of employing a dynamic update strategy in DOR, since it helps mitigate overfitting to noise (Wei et al., 2021) and achieves superior calibration performance.

### H.4. The choice of lexical database

In this work, we construct a set of textual outliers using nouns from WordNet. To evaluate whether different lexical databases significantly affect the results, we performed an ablation study on the choice of lexical databases. Specifically, we consider two additional textual databases: CLIP's vocabulary and ConceptNet 5.7. For CLIP's vocabulary, it includes 49,407 characters and words. For ConceptNet, we use raw sentences and filter out those exceeding CLIP's input limit (77 tokens). Finally. it consists of 705,662 short sentences. We report the average calibration results on the base-to-new datasets.

*Table 14.* The ablation on the repository of the outlier set. The average ECE on the base-to-new datasets is compared. DOR is insensitive to lexical databases.

| Method | Class | Vanilla | WordNet | CLIP | ConceptNet5 |
|---|---|---|---|---|---|
| CoOp | Base | 3.07 | **2.67** | 2.91 | 3.02 |
| | New | 14.58 | **6.49** | 8.43 | 8.37 |
| | HM | 8.83 | **4.58** | 5.67 | 5.70 |
| MaPLe | Base | **2.75** | 2.93 | 2.86 | 3.19 |
| | New | 5.46 | **4.71** | 5.11 | 5.24 |
| | HM | 4.11 | **3.82** | 3.99 | 4.22 |

As shown in Table 14, we find that DOR can achieve the best calibration performance with WordNet and all databases can achieve better results than the baseline. Additionally, we observed that short sentences may not perform as well as prompts like "a photo of [class]."

*Table 15.* Calibration results of ECE (%) of prompt tuning methods with DOR on pathMNIST. "Vanilla" denotes the baseline of fine-tuning methods. **Bold** numbers are significantly superior results. DOR boosts the performance of existing methods on confidence calibration.

| | ZSCLIP | CoOp | | KgCoOp | | MaPLe | | DEPT | | PromptSRC | | CoPrompt | |
|---|---|---|---|---|---|---|---|---|---|---|---|---|---|
| Class | Vanilla | Vanilla | + DOR | Vanilla | + DOR | Vanilla | + DOR | Vanilla | + DOR | Vanilla | + DOR | Vanilla | + DOR |
| Base | 29.80 | 1.56 | **1.25** | 13.45 | **12.15** | 14.12 | 15.47 | 6.57 | 7.63 | 12.43 | **11.52** | 12.26 | **6.27** |
| New | 15.27 | 61.28 | **14.99** | 12.45 | **7.48** | 13.47 | **6.54** | 62.18 | **13.91** | 11.08 | **10.34** | 8.39 | **7.57** |
| HM | 22.54 | 31.42 | **8.12** | 12.95 | **9.82** | 13.80 | **11.01** | 34.38 | **10.77** | 11.76 | **10.93** | 10.33 | **6.92** |

*Table 16.* Accuracy (%) of prompt tuning methods with DOR on pathMNIST. "Vanilla" denotes the baseline of fine-tuning methods. **Bold** numbers are significantly superior results. DOR boosts the performance of existing methods on generalization.

| | ZSCLIP | CoOp | | KgCoOp | | MaPLe | | DEPT | | PromptSRC | | CoPrompt | |
|---|---|---|---|---|---|---|---|---|---|---|---|---|---|
| Class | Vanilla | Vanilla | + DOR | Vanilla | + DOR | Vanilla | + DOR | Vanilla | + DOR | Vanilla | + DOR | Vanilla | + DOR |
| Base | 24.64 | 92.52 | **93.52** | 87.77 | 86.89 | 85.87 | 85.73 | 92.33 | 91.37 | 92.16 | 91.58 | 88.70 | 88.55 |
| New | 44.71 | 31.58 | **35.68** | 40.80 | **47.31** | 35.46 | **40.13** | 30.06 | **44.57** | 42.46 | **43.47** | 41.72 | **50.27** |
| HM | 34.68 | 62.05 | **64.60** | 64.29 | **67.10** | 60.67 | **62.93** | 61.20 | **67.97** | 67.31 | **67.53** | 65.21 | **69.41** |

## I. Comparison with existing calibration methods

To further validate the effectiveness of our proposed DOR, we compare it with two recent calibration approaches for CLIP. We consider two main calibration strategies: post-hoc scaling and regularization-based training. For post-hoc scaling, we compare DOR with Zero-Shot-Enabled Temperature Scaling (ZS-TS) (LeVine et al., 2023). Specifically, we perform post-hoc calibration on the fine-tuned model using ImageNet-1k and evaluate the learnable temperature $\tau$ on both the base and new classes. For regularization-based calibration, we incorporate DOR into calibrated robust fine-tuning method (CaRot) (Oh et al., 2024) like Equation 10. We report the average ECE performance on the base-to-new datasets.

**DOR outperforms post-hoc scaling under base-to-new evaluation.** We present the results in Table 17. We observe that the temperature $\tau$ optimized on ImageNet-1k significantly mitigates the overconfidence and improves the calibration of the

*Table 17.* Average ECE (%) comparison with post-hoc scaling methods on base-to-new datasets. ZS-TS fails to calibrate the base classes. DOR effectively mitigates the miscalibration issue on new classes while maintaining the calibration performance on the base classes.

| | CoOp | | | CoCoOp | | | KgCoOp | | | MaPLe | | | PromptSRC | | |
|---|---|---|---|---|---|---|---|---|---|---|---|---|---|---|---|
| | Vanilla | +ZS-TS | +DOR | Vanilla | +ZS-TS | +DOR | Vanilla | +ZS-TS | +DOR | Vanilla | +ZS-TS | +DOR | Vanilla | +ZS-TS | +DOR |
| base | 3.07 | 8.25 | **2.67** | **3.60** | 9.12 | 4.22 | **5.82** | 8.49 | 6.07 | **2.75** | 6.68 | 2.83 | **3.75** | 6.74 | 3.88 |
| new | 14.58 | 7.96 | **6.49** | 6.14 | 4.81 | **4.02** | 4.48 | **3.36** | 3.99 | 5.46 | **4.05** | 4.44 | 4.15 | **3.48** | 3.80 |
| HM | 8.83 | 8.11 | **4.58** | 4.87 | 6.97 | **4.12** | 5.15 | 5.93 | **5.03** | 4.11 | 5.37 | **3.64** | 3.95 | 5.11 | **3.84** |

*Table 18.* ECE (%) comparsion with regularization-based methods on base-to-new datasets. DOR can incorporate with CaRot to achieve better calibration.

| | | Caltech101 | Pets | Cars | Flowers | Food101 | FGVC | SUN397 | DTD | EuroSAT | UCF101 | ImageNet | AVG |
|---|---|---|---|---|---|---|---|---|---|---|---|---|---|
| Base | CaRot | 7.68 | 6.14 | 12.93 | 8.34 | 6.92 | 6.15 | 5.31 | 6.68 | 10.12 | 6.76 | 3.05 | 7.28 |
| | CaRot+DOR | **5.71** | **5.10** | **11.35** | 8.55 | **6.13** | **5.72** | 5.23 | 6.86 | **9.67** | 6.81 | **2.91** | **6.73** |
| New | CaRot | 2.51 | 6.46 | 4.32 | 5.66 | 7.16 | 5.20 | 3.54 | 6.03 | 6.37 | 5.09 | 1.78 | 4.92 |
| | CaRot+DOR | 2.87 | **4.93** | **3.71** | **4.66** | **6.39** | 5.74 | 3.62 | **5.10** | 8.74 | **4.92** | 1.80 | **4.77** |

fine-tuned CLIP on new classes. For instance, it reduces the Expected Calibration Error (ECE) of the state-of-the-art method PromptSRC from 4.15% to 3.48%. However, such calibration can not used on the base classes. ZS-TS exacerbates the model's underconfidence and increases the ECE from 3.75% to 6.74% on PromptSRC. In contrast, our approach effectively mitigates the miscalibration issue on new classes while maintaining the calibration performance on the base classes.

**DOR boosts existing regularization-based methods for better calibration.** As shown in Table 18, we observe that CaRot achieves decent calibration performance on both base and new classes. Furthermore, our DOR method can be effectively integrated into CaRot and improve calibration on both base and new classes. For instance, DOR reduces the ECE by 1.14% and 1.21% on base and new classes of the OxfordPets dataset, respectively. This demonstrates that DOR is a flexible regularization strategy compatible with various fine-tuning methods.

## J. Application on medical imaging

To verify our proposed DOR can be applied in real-world tasks, we conduct the experiments on PathMNIST from MedMNIST+ (Yang et al., 2023) as the medical benchmark. PathMNIST is comprised of 9 types of tissues, resulting in a multi-class classification task. For the text of each label, we use the caption from the official implementation. Specifically, the dataset includes the following labels: adipose tissue (0), background (1), debris (2), lymphocytes (3), mucus (4), smooth muscle (5), normal colon mucosa (6), cancer-associated stroma (7), and colorectal adenocarcinoma epithelium (8). we use We fine-tune the CLIP with 16 shots from the first 5 classes and evaluate the model on all 9 classes under the base-to-new evaluation protocol.

As shown in Table 15 and 16, we find that DOR can effectively help with the calibration of fine-tuned CLIP on medical datasets. Noted that existing prompt tuning methods can be effectively applied to medical image datasets. For instance, after fine-tuning, the accuracy of the base class improved significantly from 29.80% to over 85% for all methods. However, compared with standard benchmarks used in the main experiment, the calibration performance could be worse and output worse ECE for new classes. To this end, DOR effectively reduces ECE across all methods. For example, DOR lowers the ECE for new classes from 61.28% to 14.99% in CoOp. Moreover, DOR can fit existing advanced methods like CoPrompt and significantly reduces the overall ECE from 10.33% to 6.92%. In summary, DOR can notably improve the calibration performance of prompt-tuning methods and is capable of real-world domain-specific tasks.

## K. Feature visualization of DOR-V

In the discussion (Section 6), we show that DOR-V can successfully reduce the calibration error for visual feature adaptation. To investigate how DOR influences the feature space of base classes when incorporating visual outliers, we visualized the performance of CLIP-Adapter on the DTD dataset. For a better view, we randomly selected 10 base classes for visualization. We denote DOR-V to our method combined with CLIP-Adapter.

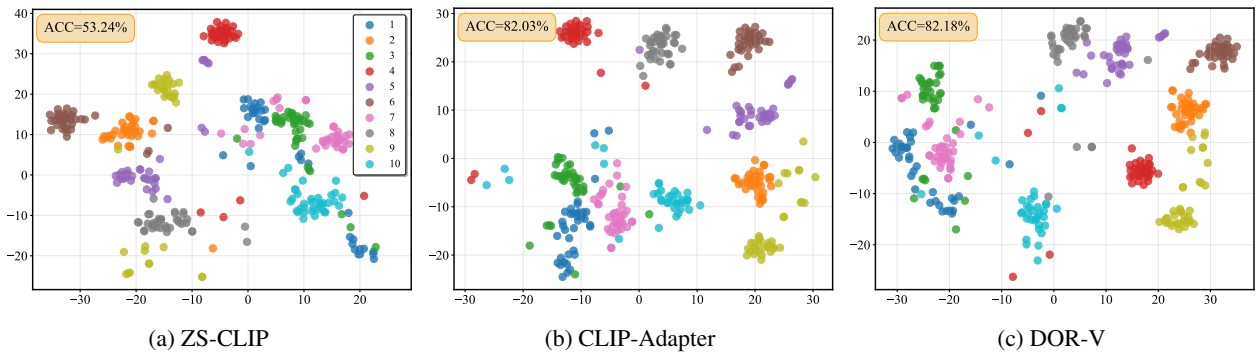

(a) ZS-CLIP          (b) CLIP-Adapter          (c) DOR-V

*Figure 11.* The t-SNE plots for visualizing visual features on the base classes of DTD dataset. DOR-V denotes our method applied on CLIP-Adapter. CLIP-Adapter generates more discriminative features compared with ZS-CLIP. and DOR does not significantly affect the visual feature space.

*Table 19.* Distribution similarity between visual features in Zero-Shot CLIP (Z), CLIP-Adapter (C), and CLIP-Adapter with DOR-V (D) on the DTD dataset.

| Metric | $Z \longleftrightarrow C$ | $Z \longleftrightarrow D$ | $C \longleftrightarrow D$ |
|---|---|---|---|
| MMD | 0.39 | 0.84 | 0.22 |
| Wasserstein | 1.48 | 1.01 | 0.47 |

We present the visualization in Figure 11. Compared to ZS-CLIP, CLIP-Adapter generates more discriminative features. Importantly, we observe that DOR does not significantly affect the visual feature space and maintains accuracy on the base class. To further quantify the difference between visual distributions, we measure the distance between distributions via Maximum Mean Discrepancy (MMD) and Wasserstein distance. As shown in Table 19, compared with ZS-CLIP, the gap between CLIP-Adapter and DOR-V is relatively smaller. These results confirm that DOR does significantly affect the feature space and can achieve better calibration results as evidenced in Table 7.

