# OpenReview forum: "Understanding and Mitigating Miscalibration in Prompt Tuning for Vision-Language Models"
_ICML.cc/2025/Conference — ICML 2025 poster_

### Official Review · Reviewer_Miv5 · 2025-03-08

**Overall Recommendation:** 3

**Summary:**

This paper investigates miscalibration issues in fine-tuned vision-language models (VLMs) like CLIP, revealing a trade-off between base and new classes where standard prompt tuning (e.g., CoOp) leads to overconfidence in new classes, while regularization-based methods (e.g., KgCoOp) cause underconfidence in base classes​
. To address this, the authors propose Dynamic Outlier Regularization (DOR), which samples textual outliers from WordNet and minimizes their feature deviation before and after fine-tuning, preventing excessive divergence in new class representations while maintaining base class accuracy​
. Extensive experiments across multiple datasets demonstrate that DOR significantly improves calibration without compromising model performance, and the technique can also be extended to visual fine-tuning methods​.

**Claims And Evidence:**

The claims made in the paper are **generally well-supported** by evidence:

**1. Identification of Calibration Trade-Off in Fine-Tuned CLIP**
Claim: Fine-tuning CLIP with prompt tuning (e.g., CoOp) leads to overconfidence in new classes, whereas regularization-based tuning (e.g., KgCoOp) results in underconfidence in base classes.
Evidence:
Empirical Analysis (Section 3.1 & 3.2): The authors analyze textual feature divergence (FD score) and show that CoOp increases FD, causing overconfidence in new classes, while KgCoOp reduces FD, leading to underconfidence in base classes.

**2. Effectiveness of Dynamic Outlier Regularization (DOR)**
Claim: DOR mitigates the calibration trade-off by sampling textual outliers from WordNet and minimizing their feature deviation, improving both base and new class calibration.
Evidence:
Table 1 & Table 2: Show that DOR consistently reduces ECE across 11 datasets when applied to various tuning methods (CoOp, CoCoOp, MaPLe, etc.).

**3. Generalization of DOR to Visual Fine-Tuning (DOR-V)**
Claim: DOR can be extended to visual fine-tuning by applying a similar regularization to image representations (DOR-V).
Evidence:
Table 5: Shows that DOR-V improves calibration in visual fine-tuning methods (VPT and CLIP-adapter) across multiple datasets.

**Essential References Not Discussed:**

Not Discovered.

**Experimental Designs Or Analyses:**

The authors conducted evaluations using the **two standard evaluation protocols**, **Base-to-New Generalization** and **Domain Generalization**, while comparing against **comprehensive and recent baselines**, such as **MaPLe and DEPT**. The **calibration metrics** used are also comprehensive, overall, the **experimental design and analysis are effective**.

**Methods And Evaluation Criteria:**

Basically, the authors sample new outliers to increase the diversity of text prompts (without making them exactly the same as the original prompts). This introduces an appropriate constraint during model optimization, enabling the fine-tuned model to **reduce overconfidence on new classes** when encountering them, thereby enhancing its generalization ability. Overall, the approach **makes sense**.

**Other Comments Or Suggestions:**

None

**Other Strengths And Weaknesses:**

Clear motivation, reasonable methodology, effective experiments, a well-structured paper.

**Questions For Authors:**

In your experiments, you mentioned that **randomly selecting outliers** also provides a **strong baseline**. Does this suggest a **valuable insight** for real-world **OOD applications**? How do you evaluate the improvement in **generalization performance** between **Near-Outliers and Random-Outliers** in practical applications?

**Relation To Broader Scientific Literature:**

Miscalibration is an overlooked problem in the field of **prompt tuning**, and this work proposes an effective method to address it.

**Theoretical Claims:**

I have checked **Appendix C. Theoretical Justification**, and it looks correct.

---

> ### Author Rebuttal · Authors · 2025-04-01
>
> Thank you for your insightful and positive suggestions. Here’s our response below:
>
> ### 1. The choice of Near-Outliers or Random-Outliers
>
> We agree that random outliers offer a practical and efficient approach for real-world out-of-distribution (OOD) applications. To investigate their feasibility, we conduct a detailed analysis of when random outliers are sufficiently effective. We hypothesize that the performance gap between near OOD and random OOD may depend on the task’s scope. Accordingly, we compare them across datasets of varying breadth: ImageNet, spanning diverse object classes, versus the fine-grained DTD (textures) and Flowers (flowers). In the table below, our results reveal a larger gap between random and near OOD on ImageNet compared to the two fine-grained datasets, suggesting that random outliers may be particularly viable for tasks with broad and diverse categories in our method.
>
>
>  | Method  | DTD   | Flowers | ImageNet |
> |---------|-------|---------|----------|
> | Near    | 8.58  | 6.12    | 1.64     |
> | Random  | 9.54  | 6.61    | 1.66     |
> | Δ(Gap)  | 0.96  | 0.49    | 0.02     |

---

> > ### Comment · Reviewer_Miv5 · 2025-04-01
> >
> > Thanks for the response. I do not have any further questions and support the acceptance of this paper.

---

> > > ### Author Response · Authors · 2025-04-08
> > >
> > > Dear Reviewer Miv5,
> > >
> > > Thank you for supporting the acceptance of this paper. We're glad that our responses addressed all the concerns. We truly appreciate your valuable time for the reviewing.
> > >
> > > Best regards,
> > >
> > > Authors of submission 5881

---

### Official Review · Reviewer_MfZm · 2025-03-13

**Overall Recommendation:** 3

**Summary:**

This paper identifies a calibration trade-off in existing prompt tuning methods: standard tuning (e.g., CoOp) leads to overconfidence on new classes due to increased textual label divergence, while regularization-based tuning (e.g., KgCoOp) results in underconfidence on base classes despite improved accuracy. To address this, the authors propose Dynamic Outlier Regularization (DOR), a method that regulates textual divergence of novel classes using dynamically sampled textual outliers from a large vocabulary (e.g., WordNet), without restricting base class features.

**Claims And Evidence:**

The primary claim—that existing prompt tuning methods compromise calibration on either base or new classes—is backed by empirical analysis on datasets like StanfordCars and UCF101, showing CoOp’s overconfidence (ECE 14.58% on new classes) and KgCoOp’s underconfidence (ECE 5.82% on base classes). The explanation via textual feature divergence is substantiated with Feature Divergence (FD) scores and logit gap visualizations (e.g., Figures 2 and 3). The claim that DOR mitigates this trade-off is supported by comprehensive results.

**Essential References Not Discussed:**

N/A

**Experimental Designs Or Analyses:**

I reviewed the experimental designs and analyses, focusing on Sections 5.1-5.2 and Appendices F-J. The results are convincing.

**Methods And Evaluation Criteria:**

The proposed DOR method—minimizing textual feature discrepancy of dynamically sampled outliers—is sensible for addressing calibration in VLMs, as it targets the identified cause (textual divergence) without altering core fine-tuning objectives.

**Other Comments Or Suggestions:**

N/A

**Other Strengths And Weaknesses:**

Strength
- The paper is original in combining dynamic outlier regularization with prompt tuning to break the calibration trade-off, a novel synthesis of ideas.
- Its significance lies in improving VLM reliability for safety-critical applications (e.g., medical diagnosis), and the clarity of writing, figures (e.g., Figure 1), and tables enhance accessibility. T
- he extensive evaluation across 11 datasets and 4 ImageNet variants is a major strength.

Weakness
- The theoretical analysis is limited to binary classification, reducing its generalizability to multi-class settings.
- Method shows minor improvements on some of the dataset
- Results do not include standard deviation over multiple runs, making it difficult to understand the significance of the improvement

**Questions For Authors:**

- The in Appendix C assumes binary classification. How do the authors expect textual divergence to influence confidence in multi-class settings with more complex logit distributions?
- Table 10 shows Euclidean distance slightly outperforms cosine similarity. Why do the authors prefer cosine as the default, beyond convention? An explanation (e.g., stability, alignment with CLIP’s training) could strengthen the method’s justification.

**Relation To Broader Scientific Literature:**

The paper builds on prior work in VLM fine-tuning and calibration. It extends CoOp, KgCoOp and more by addressing their calibration limitations, aligning with findings in existing works on novel class miscalibration.

**Theoretical Claims:**

The paper includes a theoretical justification in Appendix C, linking textual divergence to confidence via logit variance in a binary classification setting (Proposition C.1). I checked the proof, which assumes logits follow a normal distribution and shows that higher variance (σ) increases expected maximum softmax probability.

---

> ### Author Rebuttal · Authors · 2025-04-01
>
> Thanks for your helpful and positive feedback. Our response is below:
>
> ### 1. Theoretical analysis on multi-class settings
>
> Thank you for the insightful suggestion. We presented the theoretical results in binary classification to help readers quickly understand the relationship between feature divergence and output confidence. Here, we draft a theoretical analysis in the multi-class setting. In particular, a set of logits $\\{z_j\\}_ {j=1}^K$ sampled from a Gaussian distribution $\mathcal{N}(\mu, \sigma^2)$ in $K$-class classification and maximum softmax probability is denoted by $p_{\max} = \max_j \frac{e^{z_j}}{\sum_k e^{z_k}}$. Now, we prove that the expected maximum softmax probability $\mathbb{E}[p_{\max}]$ increases strictly with $\sigma$.
>
> Similar to our former analysis, we assume $\mu = 0$ without loss of generality. We then standardized the logits as $z_j = \sigma u_j$, with $u_j \sim \mathcal{N}(0, 1)$ i.i.d., so $p_{\max} = \max_{1 \leq j \leq K} \frac{e^{\sigma u_j}}{\sum_{k=1}^K e^{\sigma u_k}}$. To analyze the monotonicity of $\mathbb{E}[p_{\max}]$ with respect to $\sigma$, we compute the derivative $\frac{d}{d\sigma} \mathbb{E}[p_{\max}]$. By using the Dominated Convergence Theorem, we interchange differentiation and expectation. The derivative simplifies to $\frac{d}{d\sigma} \mathbb{E}[p_{\max}] = \mathbb{E} \left[ p_m \left( u_m - \sum_{k=1}^K p_k u_k \right) \right]$, where $m = \arg\max_j p_j$. We ensure this step is valid by bounding the derivative $\left| \frac{\partial p_m}{\partial \sigma} \right| \leq 2 \max_j |u_j|$, which has a finite expectation for fixed $K$. Now, since $u_m = \max_j u_j$ and $e^{\sigma u_j}$ is increasing in $u_j$, we have $u_m - \sum_{k=1}^K p_k u_k = \sum_{k \neq m} p_k (u_m - u_k)$. Since $u_m > u_k$ for $k \neq m$ almost surely, and $\sum_{k \neq m} p_k = 1 - p_m > 0$, this expression is strictly positive. Given $p_m > 0$, the expectation $\mathbb{E} \left[ p_m \left( u_m - \sum_{k=1}^K p_k u_k \right) \right]$ is positive, implying $\frac{d}{d\sigma} \mathbb{E}[p_{\max}] > 0$. Hence, for $\sigma_2 > \sigma_1$, we conclude $\mathbb{E}[p_{\sigma_2}] > \mathbb{E}[p_{\sigma_1}]$ in the multi-class setting. We will incorporate this detailed analysis into the final manuscript.
>
>
>
>
> ### 2. Minor improvements on some datasets
>
> Thank you for raising the concern. We also identify that the severity of the calibration problem differs on various datasets: current prompt-tuning methods are miscalibrated on some datasets (e.g., DTD, EuroSAT, and FGVCAircraft) and well-calibrated on some other datasets (e.g., Caltech101 and ImageNet). We conjecture that the calibration performance might be relevant to the distribution distances between the pretraining dataset of CLIP and downstream datasets (Larger distances may cause worse calibration). Notably, our method can consistently improve the calibration performance on various datasets, especially significant on those challenging cases.
>
>
>
>
> ### 3. Add standard deviation to results
>
> Thanks for your valuable suggestion. We present the main results with standard deviation in Table 1 of [[link](https://anonymous.4open.science/r/icml_rebuttal-5881/rebuttal.pdf)]. The results confirm the significance of the improvement from our method.
>
>
> ### 4. The choice of cosine similarity
>
> We use the cosine similarity since it is widely adopted in the literature of CLIP. In addition, the features of CLIP are typically normalized so that using either Euclidean distance or cosine distance achieves similar impacts. In particular, the squared Euclidean distance is proportional to the cosine distance: $||\mathbf{x} - \mathbf{y}||_2^2 = 2 - 2 \cos \langle \mathbf{x}, \mathbf{y} \rangle$. Therefore, we default to using the cosine similarity due to its popularity in vision-language models.

---

> > ### Comment · Reviewer_MfZm · 2025-04-01
> >
> > I would like to thank the authors for taking the time to respond to the questions. After reviewing the rebuttal and considering comments from other reviewers, I maintain my decision in favour of acceptance.

---

> > > ### Author Response · Authors · 2025-04-08
> > >
> > > Dear Reviewer MfZm,
> > >
> > > Thank you for supporting the acceptance of this work. We truly appreciate your valuable reviews and suggestions.
> > >
> > > Best regards,
> > >
> > > Authors of submission 5881

---

### Official Review · Reviewer_QPox · 2025-03-13

**Overall Recommendation:** 3

**Summary:**

1. This paper analyzes the trade-off between base and novel classes from the perspective of textual distribution divergence.

2. This paper proposes a simple DOR regularization method, which can compile with existing prompt learning methods.

3. Experimental results show promising performance compared with related methods.


## update after rebuttal
Thanks for the author's responses, and I will maintain my original score.

**Claims And Evidence:**

The claims about the calibrating are basically supported by experimental results and theoretical analysis.

**Essential References Not Discussed:**

More recent prompt learning methods should be analyzed and discussed, such as "Gallop" "PromptKD" and more.

**Experimental Designs Or Analyses:**

The experimental design is basically reasonable.

**Methods And Evaluation Criteria:**

The used benchmark datasets are reasonable in prompt learning fields.

**Other Comments Or Suggestions:**

None.

**Other Strengths And Weaknesses:**

Strengths:

1. The idea of calibration is reasonable, which has not been explored enough in the field of prompt learning before.

2. This paper gives the theoretical analysis from feature divergence to miscalibration along with the trade-off between base and novel classes.

3. The designed method is simple, and the results are promising.

Weaknesses:

Please refer to "Questions For Authors".

**Questions For Authors:**

1.  In my opinion, this paper uses the extra-textual information to mimic the new classes, through the similarity connection and existing word datasets. So, maybe it is not fair to compare with other methods directly to some degree, considering the "information leakage". This paper should consider this part carefully.

2. More recent prompt learning methods, such as Gallop and PromptKD, should be equipped with DOR, to show the performance. In fact, PromptKD also involves extra information, so the performance of PromptKD with DOR will give some insights and discussions.

3. The performance change should be discussed. As shown in Table 3, DOR decreases the base classes' performance in some methods, and what are the reasons?

4. Some visualization results about outliers or methods with/without DOR are helpful for understanding the effect of the proposed method.

**Relation To Broader Scientific Literature:**

This paper gives insight into the miscalibration in existing prompt learning methods, which has not been discussed and digged before.
Meanwhile, the calibration idea can be further explored in this field, so I think it contributes to the broader scientific literature to some degree.

**Theoretical Claims:**

This paper provides a theoretical analysis of the feature divergence and miscalibration, and gives the theoretical justification in the supplementary material.

---

> ### Author Rebuttal · Authors · 2025-04-01
>
> ### 1. Clarification on "information leakage"
>
> Thanks for your insightful review. We believe this concern is closely related to the response #5 for Reviewer YQTf, which is about the overlap between outliers and new classes. We'd like to clarify that our method does not leak the information of new classes, as the outliers are selected with base classes. Moreover, the ablation study in Table 6 demonstrates that using even far-OOD and random-OOD can significantly improve the overall calibration performance. In the response #5 for Reviewer YQTf, we also provide an additional ablation study by removing the new classes from the outlier pool (e.g., WordNet). The results demonstrate that the effectiveness of our method does not rely on the overlap with new classes. Instead of causing "information leakage", our work opens up the possibility of utilizing cheap public texts for improving calibration performance. In summary, leverages general semantic information from language space rather than memorizing target classes, which ensures its fairness and generalizability. We will add this discussion to the final version.
>
>
>
> ### 2. Results on PromptKD
>
> Thank you for the great suggestion. we provide new results by incorporating DOR into a recent method - PromptKD. In particular, we apply DOR in the first stage to train a large teacher model(ViT-L-14), and keep the second stage unchanged. The results below show that our method can achieve meaningful improvements over PromptKD despite its strong performance.
>
> | Method   | Base       | New        | HM          |
> |----------|------------|------------|-------------|
> | PromptKD | 4.73±0.36  | 4.38±0.53  | 4.56±0.45   |
> | +DOR     | 4.81±0.23  | **3.66±0.43**  | **4.24±0.33**   |
>
>
> ### 3. Performance decrease on base classes
>
> Thank you for the careful review. We'd like to clarify that our method achieves comparable accuracy to baselines on base classes. To illustrate this, we provide the accuracy with standard deviation in the table below. In particular, the performance gaps are negligible with most of them are less than 0.2%. We conjecture that adding regularization may slightly affect the training dynamic during the prompt tuning, leading to trivial changes on the accuracy (also increasing a little on new classes). Overall, our method can maintain the generalization performance of baselines on Base classes.
>
>
> |         | CoOp      | CoCoOp    | KgCoOp    | MaPLe     | DEPT      | TCP       | CoPrompt  | PromptSRC | PromptKD  |
> |---------|-----------|-----------|-----------|-----------|-----------|-----------|-----------|-----------|-----------|
> | Vanilla | 82.97±0.56| 80.57±0.47| 82.29±0.25| 82.11±0.54| 83.70±0.32| 83.95±0.40| 82.32±0.51| 84.77±0.29| 85.74±0.35|
> | +DOR    | 83.20±0.47| 79.89±0.57| 82.13±0.35| 82.08±0.50| 83.81±0.47| 83.89±0.29| 82.39±0.57| 84.79±0.34| 85.52±0.36|
>
> ### 4. Visualization on DOR
>
>
> Thank you for the great suggestion. we provide new visualization in Figures 1-2 [[link](https://anonymous.4open.science/r/icml_rebuttal-5881/rebuttal.pdf)]. The results show that KgCoOp significantly decreases the FD scores on both Base and New classes, whereas our method primarily impacts New classes. Consequently, our approach provides confidence scores that align with the improved accuracy on Base classes. We believe the visualization suggested by the reviewer can enhance the clarity of this paper.

---

### Official Review · Reviewer_YQTf · 2025-03-14

**Overall Recommendation:** 3

**Summary:**

The authors propose a method called Dynamic Outlier Regularization (DOR) to improve confidence calibration for both base and new classes after fine-tuning. Extensive experiments are conducted on multiple benchmark datasets to evaluate its effectiveness.

**Claims And Evidence:**

One of the major claims is that DOR outperforms kGCoOp due to its ability to prevent the increase of textual divergence on new classes. To strengthen this claim, I suggest visualizing DOR in the same manner as Figure 2 and Figure 6 in the supplementary material.
This would better illustrate how DOR improves on textual divergence and solidify the contribution.

**Essential References Not Discussed:**

N/A

**Experimental Designs Or Analyses:**

Please see Methods And Evaluation.

**Methods And Evaluation Criteria:**

I am concerned about the fair comparison in the experimental demonstration:

1. I noticed inconsistencies between the baseline results reported in the manuscript and those in [1]. For example, on new classes, in [1] CoOp reports ECE - 13.84, ACE - 13.76, MCE - 3.80, PIECE - 14.71; while in this paper they are ECE -14.58, ACE - 14.50, MCE - 3.73, PIECE - 15.27.
Can the authors clarify why these numbers differ? If not justified, this could affect the validity of the comparison.

2. The authors claim that DOR improves over kGCoOp by preventing textual divergence from increasing on new classes. To this end, to ensure a fair comparison, I believe the manuscript should compare various baselines+DOR vs. various baselines+kGCoOp, as done in Table 2 & 3 of [1].


[1] Wang et al., Open-Vocabulary Calibration for Fine-tuned CLIP, ICML 2024

**Other Comments Or Suggestions:**

N/A

**Other Strengths And Weaknesses:**

I find the overall presentation flow to be clear and easy to follow. The method is also well-described.

The core idea is interesting, but it lacks solid support in certain areas, as I mentioned in Claims and Evidence & Methods and Evaluation Criteria. Strengthening these aspects would definitely enhance the technical contribution.

**Questions For Authors:**

One of the core components of the proposed method is the choice of textual outliers in Equation (7). I believe more discussion and experimentation on this part is necessary.

Ablation study on outlier set size:

(1) Do the authors have an ablation study on how the size of the outlier set affects performance? Is the method sensitive to the outlier set size?

(2) How similar is the sampled outlier set to the actual new classes? Because I feel like one possible reason this method is effective might be that the sampled outliers overlap with or are similar to new classes.

I am open to raising my score if the authors can address these concerns.

**Relation To Broader Scientific Literature:**

The authors provide a good coverage of the related work.

**Theoretical Claims:**

I think the theoretical claims are sound to me.

---

> ### Author Rebuttal · Authors · 2025-04-01
>
> Thank you for the constructive feedback. Please find our response below:
>
>
> ### 1. Visualization of DOR
> Thanks for the great suggestion. We'd like to clarify that our method outperforms KgCoOp as our method does not fix the confidence level on the base classes while preventing the increase of textual divergence on new classes. In contrast, KgCoOp anchors the confidence level, leading to underconfidence on Base classes. To illustrate this, we provide new visualization in Figures 1-2 [[link](https://anonymous.4open.science/r/icml_rebuttal-5881/rebuttal.pdf)]. The results show that KgCoOp significantly decreases the FD scores on both Base and New classes, whereas our method primarily impacts New classes. Consequently, our approach provides confidence scores that align with the improved accuracy on Base classes. We believe the visualization suggested by the reviewer can enhance the clarity of this paper.
>
>
> ### 2. Clarification on result inconsistency
>
> Thank you for your thorough review. We have carefully revisited the CoOp performance reported in our manuscript and DAC. We found that the slight differences occur specifically on two datasets—FGVCAircraft and EuroST—while the results for the remaining nine datasets align closely with those in DAC. We attribute these minor variations to differences in experimental settings, such as the random seed, hardware specifications, or software environment, which can subtly affect model performance in certain cases. Notably, these minor variations do not impact the contribution of DOR, which largely improves the calibration performance. For reproduction, we will release the full code of baselines on GitHub after publication.
>
>
>
> ### 3. Comparison between DOR and KG
>
> Thank you for the great suggestion. We conduct new experiments by comparing DOR and KgCoOp on 3 baselines (CoOp, MaPLe, and CoPrompt), and present the results in the table below. The results show that integrating with our method can **consistently outperform** those with KgCoOp on overall performance. In particular, our method achieves much better performance than KgCoOp on Base classes, while KgCoOp performs well on New classes. This is consistent with the analysis presented in Subsection 3.2: KgCoOp anchors the confidence level, leading to underconfidence on Base classes. In short, the results demonstrate the superiority of DOR in breaking the calibration trade-off between base and novel classes.
>
> |          | Method  | Base | New  | HM    |
> |----------|---------|------|------|-------|
> | **CoOp**     | Vanilla | 3.07 | 14.49 | 8.78  |
> |          | +KG     | 5.82 | 4.48  | 5.15  |
> |          | +DOR    | 2.47 | 6.48  | **4.47** |
> | **MaPLe**    | Vanilla | 2.75 | 5.46  | 4.11  |
> |          | +KG     | 4.01 | 4.29  | 4.15  |
> |          | +DOR    | 3.06 | 4.26  | **3.66** |
> | **CoPrompt** | Vanilla | 2.60 | 5.96  | 4.28  |
> |          | +KG     | 4.01 | 4.99  | 4.50  |
> |          | +DOR    | 2.98 | 5.14  | **4.06** |
>
>
>
> ### 4. Ablation on the size of the outlier set
>
> Yes, we present the ablation study in `Appendix H2`. The results show that our method is not sensitive to the outlier set size, especially when the size is larger than 500. In particular, the impact of set size is negligible on Base classes, while a larger set size promotes better performance on New classes. In an extreme case with $k = 10$, our method can still significantly improve the performance on new classes. Due to the low cost of outlier texts, we recommend $K=5000$ as the default to achieve optimal calibration performance.
>
>
>
> ### 5. Overlap between outlier texts and new classes
>
> Thank you for raising the concern. In Table 8, we present all selected outlier texts for 6 datasets, showing almost no class overlap especially on StanfordCars and FGVCAircraft. Moreover, the ablation study in Table 6 demonstrates that using even far-OOD and random-OOD can significantly improve the overall calibration performance. To further analyze the impact of overlap, we conduct a new ablation study by excluding all new classes from the outlier pool (denoted as DOR$^\dagger$). The average results on 11 datasets below show that DOR$^\dagger$ without overlap achieves comparable performance to DOR, significantly improving the performance on all three baselines. **Therefore, the effectiveness of our method does not rely on the overlap with new classes**.
>
>
>
> | Method       | Variant       | Base  | New   | HM    |
> |--------------|---------------|-------|-------|-------|
> | **CoOp**     | Vanilla       | 3.07  | 14.58 | 8.83  |
> |              | +DOR          | 2.67  | 6.49  | 4.58 |
> |              | +DOR$^\dagger$| 2.82  | 6.77  | 4.80  |
> | **MaPLe**    | Vanilla       | 2.75  | 5.46  | 4.11  |
> |              | +DOR          | 2.83  | 4.44  | 3.64 |
> |              | +DOR$^\dagger$| 2.89  | 4.51  | 3.70  |
> | **CoPrompt** | Vanilla       | 2.56  | 5.96  | 4.26  |
> |              | +DOR          | 2.96  | 4.69  | 3.83  |
> |              | +DOR$^\dagger$| 2.71  | 4.93  | 3.82 |

---

> > ### Comment · Reviewer_YQTf · 2025-04-08
> >
> > Thank you for addressing my questions. Based on the responses, I am raising my score and now lean more towards acceptance.

---

> > > ### Author Response · Authors · 2025-04-08
> > >
> > > Dear Reviewer Knmp,
> > >
> > > Thank you for raising the score and supporting our work. We're glad to hear that all concerns have been addressed during rebuttal.  Your valuable suggestions have significantly enhanced the quality of this paper.
> > >
> > > Best regards,
> > >
> > > Authors of submission 5881

---

### Decision · Program_Chairs · 2025-05-01

**Decision:**

Accept (poster)

**Comment:**

The paper received four weak accepts. In the pre-rebuttal phase, the reviewers raised following points:

- fair comparison in some cases
- integration with more recent prompt learning methods
- possible information leakage
- theoretical analyses only for single class settings
- small improvement in some instances.

After the rebuttal stage, reviewers acknowledged the rebuttal and agree that their concerns have been resolved adequately. The post-rebuttal discussion left no major concerns. Some reviewers also appreciated the possible broader significance of the work. Therefore, the decision is to recommend acceptance of the paper. Authors are encouraged to include important comments from reviewers in the final verision.